# The *Deinococcus* protease PprI senses DNA damage by directly interacting with single-stranded DNA

Huizhi Lu [1,5], Zijing Chen[1,5], Teng Xie[1,2], Shitong Zhong[1], Shasha Suo[1], Shuang Song [1], Liangyan Wang[1], Hong Xu[1,3], Bing Tian [1,3], Ye Zhao [1,3] ✉, Ruhong Zhou [1,2,3,4] ✉ & Yuejin Hua [1,3] ✉

Bacteria have evolved various response systems to adapt to environmental stress. A protease-based derepression mechanism in response to DNA damage was characterized in *Deinococcus*, which is controlled by the specific cleavage of repressor DdrO by metallopeptidase PprI (also called IrrE). Despite the efforts to document the biochemical, physiological, and downstream regulation of PprI-DdrO, the upstream regulatory signal activating this system remains unclear. Here, we show that single-stranded DNA physically interacts with PprI protease, which enhances the PprI-DdrO interactions as well as the DdrO cleavage in a length-dependent manner both in vivo and in vitro. Structures of PprI, in its apo and complexed forms with single-stranded DNA, reveal two DNA-binding interfaces shaping the cleavage site. Moreover, we show that the dynamic monomer-dimer equilibrium of PprI is also important for its cleavage activity. Our data provide evidence that single-stranded DNA could serve as the signal for DNA damage sensing in the metalloprotease/repressor system in bacteria. These results also shed light on the survival and acquired drug resistance of certain bacteria under antimicrobial stress through a SOS-independent pathway.

DNA damage caused by continuous exogenous and endogenous stresses leads to instability and imperfection of the cellular genome[1]. To counter this, necessary cascade amplification and transduction of damage signals have evolved to guarantee the survival of organisms[2]. With the help of sensors detecting DNA lesions and transducers propagating DNA damage signals, effectors initiate the proper DNA damage response to enable cells to cope with DNA damage[3], which includes cell cycle checkpoint arrest, DNA repair, and apoptosis[4]. *Deinococcus* species, including *Deinococcus radiodurans*, serve as advantageous model organisms for studying bacterial adaptation because of their outstanding capabilities to tolerate extreme environmental stresses, such as high doses of ionizing radiation, UV radiation, oxidation, and

long periods of desiccation. Such extraordinary resistance comes from the highly effective DNA repair systems and cellular antioxidants. In addition to a series of highly expressed enzymatic antioxidants (e.g., catalase, peroxidase, superoxide dismutase), nonenzymatic carotenoids, and manganese ion antioxidant complex[5–9], *Deinococcus* employs various DNA damage repair pathways, including nucleotide excision repair, base excision repair, mismatch repair and homologous recombination, together with non-canonical DNA damage response (DDR) and repair proteins (e.g., RecA, DdrB, SSB, and PprA)[10–15]. Moreover, error-prone pathways such as translesion synthesis and non-homologous end joining are absent in *D. radiodurans*, which guarantees the faithful DNA replication and repair of this bacterium[7].

[1]MOE Key Laboratory of Biosystems Homeostasis & Protection, Institute of Biophysics, College of Life Sciences, Zhejiang University, Hangzhou, China. [2]Shanghai Institute for Advanced Study, Zhejiang University, Shanghai, China. [3]Cancer Center, Zhejiang University, Hangzhou, Zhejiang, China. [4]Department of Chemistry, Columbia University, New York, NY, USA. [5]These authors contributed equally: Huizhi Lu, Zijing Chen. ✉e-mail: yezhao@zju.edu.cn; rhzhou@zju.edu.cn; yjhua@zju.edu.cn

Originally characterized in *E. coli*, the SOS response is one of the most well-documented bacterial DDR systems, which can respond to DNA damage by triggering LexA autocleavage following RecA-ssDNA-ATP filaments formation[16]. Despite the existence of LexA homologous proteins, the canonical SOS response appears to be inactive in *D. radiodurans*, with RecA induction regulated by the PprI-DdrO system. The distinct DNA damage response pathway mediated by the metallopeptidase PprI (31.3 kDa for DG-PprI) and the transcription repressor protein DdrO (15.7 kDa for DG-DdrO) has been characterized and extensively studied in *Deinococcus* in recent years[17–19]. PprI is critical for the environmental adaptation of *Deinococcus* species that contributes to the regulation of DNA damage response genes (DDR genes) as well as normal metabolism[20–22]. In coordination with the transcriptional repressor DdrO, which binds to the radiation/desiccation response motif (RDRM)-containing promoters upstream of DDR genes, the specific cleavage of DdrO by PprI induces the expression of DDR proteins following DNA damage[23,24]. Moreover, biochemical and structural studies showed that the cleavage site of DdrO is in a loop region of its hydrophobic C-terminal domain, which is essential for DdrO dimer formation as well as promoter binding capability[25,26]. Interestingly, whereas most DDR-associated genes were upregulated after DNA damage, transcriptomic analysis revealed that the transcription level of *pprI* remained constant during the early, middle, and late phases of genome recovery[10]. Thus, it is tempting to speculate that in response to DNA damage PprI has to be activated through undefined mechanisms, enabling the efficient DdrO cleavage and transcriptional derepression of DDR genes. Several hypotheses attempt to identify the possible activation mechanisms of PprI, including Zinc shock[27], posttranscriptional modifications, and secondary messenger regulation[9,28–30]. However, a direct connection between DNA damage and PprI activation remains unclear.

DNA damage repair proteins, including nucleases, helicases, and polymerases, must be precisely controlled in vivo, since inappropriate activation is detrimental to the integrity of DNA[31]. Signal transduction plays a pivotal role in the proper recruitment and activation of these proteins to avoid genome instability. Knocking out *ddrO* is lethal to *D. radiodurans*, which is probably due to the complete elimination of derepression of DDR genes[32]. In this study, our cellular, biochemical, and structural analyses revealed how *Deinococcus* PprI senses DNA damage. Single-stranded DNA (ssDNA) physically bound to *Deinococcus geothermalis* PprI (termed DG-PprI for brevity) and stimulated its protease activity in a length-dependent manner. Crystal structures of apo-PprI and the PprI-ssDNA complex, together with biochemical, in vivo and in silico studies, further elucidated the activation mechanism and the dynamic monomer/dimer equilibrium of PprI. Collectively, these studies suggest that PprI per se acts as a sensor and transducer upon DNA damage, which represents an SOS-independent damage response in bacteria.

## Results

### The sulfate binding cavity of PprI

We first determined the crystal structure of full-length DG-PprI at the resolution of 2.8 Å (Rwork/Rfree=0.247/0.284) in the presence of $Mn^{2+}$, which is effective for its protease activity. The crystal data, together with the data collection and refinement statistics, are summarized in Supplementary Table 1. DG-PprI, which is comprised of 9 α-helices and 8 β-strands, contains three domains: an N-terminal zinc peptidase-like domain (residues 19-135), a helix-turn-helix domain (HTH domain, residues 136-176), and a C-terminal GAF-like domain (residues 177-277) (Supplementary Fig. 1 and 2a). The overall structure of DG-PprI could be virtually superimposed onto the previously solved *Deinococcus deserti* PprI (DD-PprI) apo structure (Protein Data Bank [PDB] ID: 3DTI)[33] with a root mean square deviation (RMSD) of 0.982Å for 187 Cα atoms (Supplementary Fig. 2b).

Despite weak electron density in apo structure, a catalytic manganese ion ($Mn^{2+}$) lies in the active site of DG-PprI, coordinated by the conserved HEXXH motif (HEISH in DG-PprI) of the N-terminal zinc peptidase-like domain (Fig. 1a). Interestingly, strong electron density was observed on top of the HTH domain of DG-PprI (Fig. 1a), which is interpreted as the sulfate ion present at 1.7 M in the

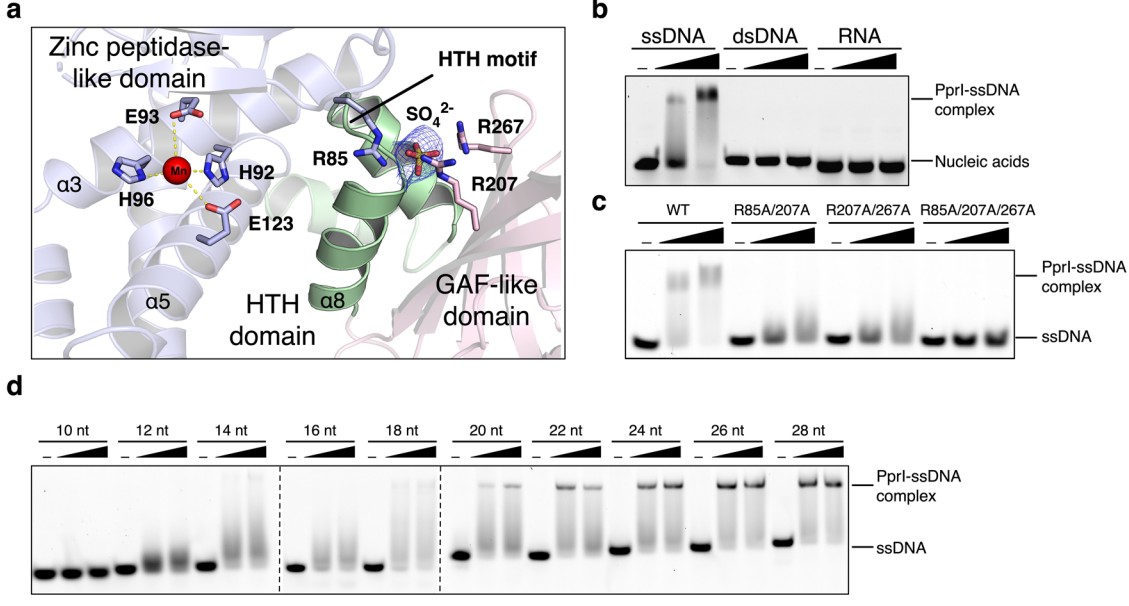

**Fig. 1 | ssDNA physically interacts with PprI. a** Close view of the sulfate and metal ion binding, with HTH motif labeled, sulfate binding residues (Arg85, Arg207, and Arg267) and HEXXH residues (His92, Glu93, and His96) shown as sticks. The electron density of sulfate is shown in blue with the refined 2Fo-Fc contoured at 1σ. **b** EMSA showing the ssDNA binding of DG-PprI. Proteins (1 or 2 μM) were incubated with 0.1 μM 5'-FAM-labeled ssDNA (35nt), dsDNA (35 bp), or RNA (35nt). **c** EMSA showing decreased ssDNA binding of the sulfate-binding cavity mutants. 35nt ssDNA (0.1 μM) was incubated with DG-PprI mutant proteins using the same reaction conditions as in panel **b**. **d** EMSA assays showing the ssDNA binding of DG-PprI in a length-dependent manner. Proteins were incubated with various length of ssDNA using the same reaction conditions as in panel **b**. Source data are provided as a Source Data file.

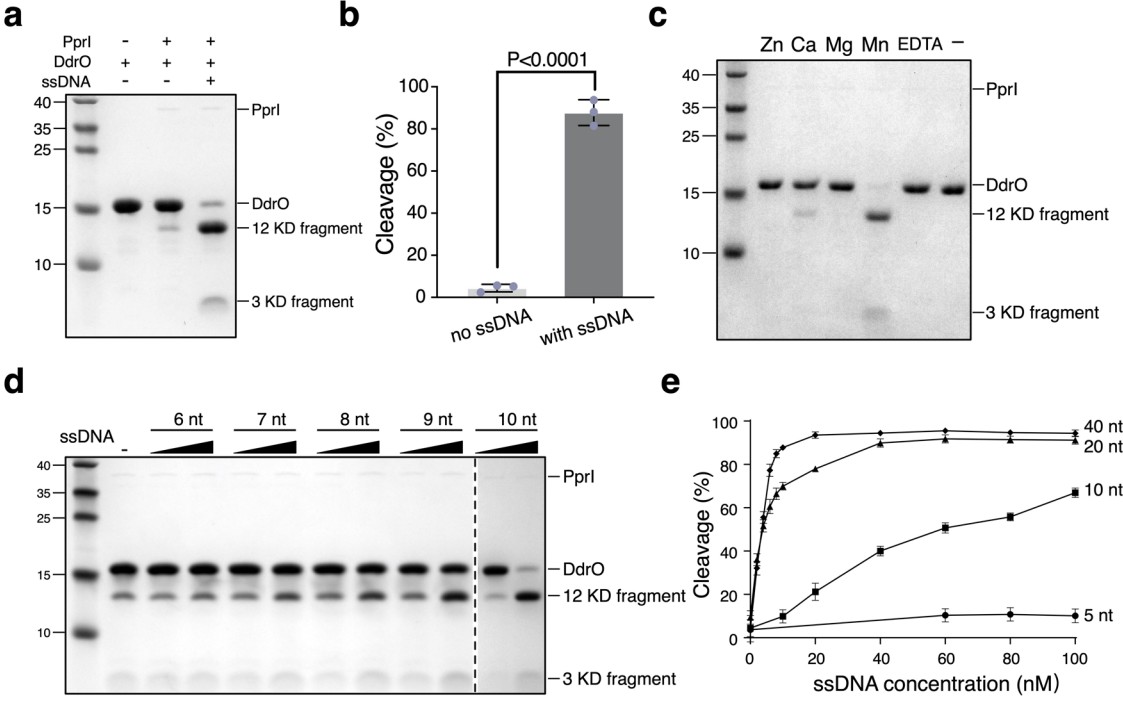

**Fig. 2 | ssDNA efficiently activates the protease activity of PprI. a** Activation assays showing the ssDNA-enhanced PprI cleavage. DG-DdrO (8 μM) was incubated with DG-PprI (0.1 μM) in the presence of 2 mM MnCl₂ in the absence or presence of 35nt ssDNA (0.1 μM) at 37 °C for 30 minutes. **b** Quantification of cleavage product of (a). Data are mean ± SD from 3 independent experiments, compared with student unpaired Student's t test (two-sided). **c** Metal ion preference of DG-PprI cleavage. DG-PprI (0.1 μM) was incubated with DG-DdrO (8 μM) and ssDNA (0.1 μM) in the presence of 2 mM EDTA or 2 mM divalent metal ions (MnCl₂, MgCl₂, CaCl₂, or ZnCl₂, respectively) at 37 °C for 30 minutes. **d** ssDNA activation assays containing various lengths (6-10nt) and concentrations (0.01 or 0.1 μM) of ssDNA using the same reaction conditions as in panel **a**. **e** Quantifications of cleavage product by various lengths (5, 10, 20, 30, 40nt) and concentrations of ssDNA using the same reaction conditions as in panel **a**. Data represent the means of the three replicates, and the bars represent their standard deviations. Source data are provided as a Source Data file.

crystallization reservoir solution. This sulfate ion is located at the positively charged cavity formed by residues from the zinc peptidase-like domain (Arg85) and GAF-like domain (Arg207 and Arg267) (Fig. 1a). Given that HTH motifs are commonly found in DNA binding proteins and sulfate might mimic the chemistry of a phosphate group, we further investigated the possible DNA binding capability of DG-PprI. In contrast to the solvent-exposed HTH motifs of well-documented transcription factors (e.g., Lex A and DdrO protein[34,35]), the HTH motif of DG-PprI is partially buried and capped by its N-terminal domain, resulting in a space suitable for ssDNA access (Fig. 1a and Supplementary Fig. 2). Electrophoretic mobility shift assays (EMSA) using ssDNA, dsDNA, and RNA were performed to further determine the possible binding activity of DG-PprI. Among all the DNA or RNA tested, DG-PprI was only able to form stable complex with ssDNA (Fig. 1b). Moreover, arginine residues forming the sulfate-binding cavity are required for ssDNA binding (Fig. 1a, c). While the double mutants (R85A/R207A and R207A/R267A) showed smeared shifted bands, the triple mutant R85A/R207A/R267A totally abolished the ssDNA binding of DG-PprI (Fig. 1c). Moreover, the ssDNA binding affinity of DG-PprI occurred in a length-dependent manner: ssDNA of 20nt constituted a length threshold, below which no stable complex was observed (Fig. 1d).

## ssDNA drastically activates the protease activity of PprI

The PprI-DdrO system is conserved within *Deinococcus* species in terms of gene regulation in vivo and biochemical properties in vitro[23–26,33]. The transcriptional level of *pprI* did not appear to be significantly upregulated during the recovery phase after DNA damage, suggesting an unknown activation mechanism of PprI cleavage in *Deinococcus*[10]. It has been shown that ssDNA can act as an upstream signal in cellular response to DNA damage[36–38]. Given the physical interactions between PprI and ssDNA (Fig. 1b, d), we tested whether the addition of ssDNA affected the protease activity of DG-PprI. Indeed, ssDNA drastically increased the cleavage of DdrO in the presence of DG-PprI even at low concentration (0.1 μM), resulting in two product fragments (3 KD and 12 KD fragments, Fig. 2a, b). Consistent with our previous findings on DG-PprI protease activity[25], the proteolytic reaction was Mn²⁺-dependent (Fig. 2c). Moreover, the stimulation of DG-PprI cleavage was DNA-length and concentration-dependent: ssDNA shorter than 8nt hardly activated DG-PprI, while longer ssDNA exhibited more effective activation (Fig. 2d, e and Supplementary Fig. 3). These results were in line with the length-dependent ssDNA-binding capability of DG-PprI (Fig. 1d). In addition, DdrO only binds RDRM-containing dsDNA specifically[25] but not ssDNA (Supplementary Fig. 4a), confirming that the DG-PprI cleavage was stimulated by the PprI-ssDNA interactions.

Because the C-terminal GAF-like domain is involved in the sulfate binding cavity formation (Arg207 and Arg267, Fig. 1a), we checked whether this domain is required for the activation of DG-PprI by ssDNA. In the absence of ssDNA, high concentration (1 μM) of GAF-like domain-truncated DG-PprI (PprI-NM) had similar cleavage activity to that of the full-length protein (Supplementary Fig. 4b), which was consistent with previous biochemical studies of DD-PprI[39]. However, PprI-NM exhibited neither ssDNA binding nor ssDNA-stimulated cleavage, in contrast to those of full-length proteins (Supplementary Fig. 4a, b). This was further confirmed by phenotypic assays showing that overexpression of PprI-NM could hardly restore the radiosensitivity of the PprI knock-out strain (Supplementary Fig. 4c), suggesting that the C-terminal GAF-like domain of PprI is critical for PprI-DdrO system activation by ssDNA.

## Interactions between PprI and ssDNA

To further investigate the detailed mechanisms of PprI activation by ssDNA, we attempted to determine the PprI-ssDNA complex structure. According to the biochemical properties of ssDNA binding of DG-PprI characterized and many co-crystallization trials with various sequences and lengths of ssDNA, crystals of the PprI-ssDNA complex containing 29nt ssDNA were grown in the presence of $Mn^{2+}$ ions and diffracted X-rays to ~4Å and subsequently improved to 2.2Å (Rwork/Rfree=0.233/0.265) (Supplementary Table 1). The overall structure of PprI-ssDNA was almost identical to the DG-PprI apo-protein structure with a slightly enlarged sulphate-binding cavity (Supplementary Fig. 5). The ssDNA is accommodated and wrapped around the hollow space between the N-terminal zinc peptidase-like domain and C-terminal GAF-domain of DG-PprI (Fig. 3a). Despite the discontinuous electron density of ssDNA due to its flexibility in crystal, we were able to build two segments of ssDNA interacting with three DNA binding patches composed of positively charged and hydrophobic residues (Fig. 3b, c and supplementary Fig. 6): (1) the sulphate-binding cavity (patch 1) and the N-terminal α1 helix of the zinc peptidase-like domain (patch 2) interacting with the 5′-GCAGTT; and (2) the C-terminal portion of ssDNA (3′-TTTTT) is anchored to the C-terminal GAF-domain of DG-PprI (patch 3). Notably, despite the weak electron density of bases of dG(7) and dT(11), dG(10) exhibited well-defined electron density with its guanine base forming π-stacking interactions with the phenylalanine residue (Phe88, Fig. 3c).

We further confirmed whether these DNA-binding patches were required for the DNA binding and activation capabilities of DG-PprI by alanine substitutions of potential ssDNA-interacting residues (Patch 1-3). Mutations of individual residues of patch 1-3 had effect on the ssDNA-binding and activation of DG-PprI to a certain extend (Supplementary Fig. 7 and 8). However, the triple mutant R85A/R207A/R267A (Patch 1 mutant) completely lost its ssDNA-binding and activation capabilities (Fig. 3c,d), indicating an essential role of patch 1 region. In contrast, the triple mutant L22A/K26A/R117A of patch 2 exhibited weakened ssDNA binding capability (smear shifted bands) but retained ssDNA activation (Fig. 3c, d, Patch 2). Interestingly, the triple mutant R220A/E250A/S251A of patch 3 was defective for ssDNA binding but could still be partially activated by ssDNA (Fig. 3c, d, Patch 3). Given the ssDNA direction built in PprI-ssDNA structure (Fig. 3a), we considered it plausible that patch 3 located at C-terminal GAF-domain possibly plays an assistant role for ssDNA capture by patch 1 region, which is consistent with the minimal length of ssDNA required for DG-PprI activation (Fig. 2d).

To verify the observed PprI-ssDNA interactions, we conducted phenotypic assays comparing the triple mutants of the ssDNA-binding patches with the wild-type DG-PprI (Fig. 3e). While overexpression of full-length *dg pprI* (YR1-*dg pprI*) fully compensated for the phenotype of the *dr_pprI* knock-out strain (YR1), the triple mutant of the patch 1 complementary strain (YR1-patch1) showed extreme sensitivity after gamma radiation treatments. The residual resistance of the YR1-patch1 strain ($10^0$ and $10^{-1}$ dilutions) may be due to the non-activated rudimentary cleavage activity of PprI in vivo. In contrast, both patch 2 (YR1-patch2) and patch 3 (YR1-patch3) mutants exhibited similar radiation resistance as the wild-type (R1) and YR1-*dg pprI* strains. Time-course qRT-PCR following gamma radiation treatments was performed to measure the transcriptional levels of representative DDR genes (*recA*, *uvrD*, and *ddrO*) regulated by the PprI-DdrO system in vivo (Fig. 3f). All three genes exhibited similar early-response patterns in both wild-type (R1) and YR1-*dg pprI* strains as previous RNA-seq results[10], which reached their peaks at the early recovery period and disappeared after 3 h upon gamma radiation treatments. However, such transcriptional inductions were not observed in the YR1-patch1 complementary strain, which exhibited less than 1.5-fold induction after irradiation. These results were consistent with the biochemical and phenotypical analyses of PprI-

ssDNA interactions (Fig. 3c, d, e), which indicated that the patch 1 interface of PprI-ssDNA plays an essential role in PprI-DdrO activation by ssDNA.

## ssDNA enhances the interaction between PprI and DdrO

Superposition of the PprI-ssDNA structure with the PprI apo-protein structure revealed almost identical overall conformation, indicating that the activation of PprI by ssDNA may not be directly induced by allosteric conformational changes of PprI (Supplementary Fig. 5). Given that the binding sites of ssDNA (patch 1 and 2) are close to the cleavage site of PprI (Fig. 3a), we suspected that ssDNA may be involved in the PprI interactions with DdrO. To test our hypothesis, PprI and DdrO proteins fused with N-terminal eYFP or eCFP, respectively, were purified and mixed at a 1:1 molar ratio to determine their interactions using fluorescence resonance energy transfer (FRET) assays (Fig. 4a). The excitation wavelength was set to 440 nm to meet the excitation wavelength requirement of eCFP-DdrO, and the emission data were collected from 460 to 600 nm. The relative fluorescence unit (RFU) ratio of 530/480 nm was calculated as the indicator of transfer efficiency to evaluate the distance between eYFP-PprI and eCFP-DdrO. While the basal RFU ratio of free eCFP and eYFP was not changed, the addition of ssDNA significantly increased the RFU ratio of the eYFP-PprI/eCFP-DdrO (Fig. 4a), suggesting that in the presence of ssDNA, PprI and DdrO were more inclined to interact with each other. This result was consistent with the ssDNA-enhanced DdrO cleavage by PprI in vitro (Fig. 2a).

We next tried to determine the crystal structure of catalytically inactive PprI complexed with DdrO and ssDNA. However, we were unable to grow crystals after multiple rounds of screening. Thus, to obtain possible structural information of ssDNA-mediated activation of DdrO cleavage by PprI, AlphaFold2 complex modeling tool[40,41] was used to predict the model of PprI binding to the C-terminal region of DdrO (full-length monomeric DdrO, Supplementary Table 4, Supplementary Fig. 9) and then DdrO dimer (Protein Data Bank [PDB] ID: 6JQ1)[25] was docked onto the predicted complex (binary complex), followed by overlaying this model on the PprI-ssDNA structure to obtain a ternary model (Fig. 4b). Furthermore, all-atom molecular dynamics (MD) simulations were performed to validate the ternary complex models, following similar protocols in our previous studies on the protein dynamics[42–46].

It has been previously reported that the cleavage site region (CSR) loop (residues 116-121 in DG-DdrO) on the C-terminal domain of DdrO is critical for the specific cleavage by PprI[24]. In the complex model, the CSR loop interacts with the N-terminal zinc peptidase-like domain of PprI by forming anti-parallel β-strands (Fig. 4b, c), and its binding stability is verified by the RMSD values (less than 2 Å) during MD simulations (Fig. 4d). Notably, the CSR loop protrudes into the cleavage site of PprI and Arg118 of the CSR loop directly interacts with the phosphate group of the ssDNA in the predicted PprI-DdrO-ssDNA model (Fig. 4b, c). Such interactions were consistent with subsequent FRET assay results using R118A mutant DdrO, which exhibited lower interaction between PprI and DdrO compared to wildtype PprI and non-enhanced PprI-DdrO interactions in the presence of ssDNA (Fig. 4a). In addition, we performed in silico mutagenesis (free energy perturbation calculations (FEP)[47–49] for R118A) to estimate the energy contribution of R118 to the binding of DdrO to PprI in the ternary model. The FEP results of R118A mutation ($\Delta\Delta G = 5.22 \pm 0.33$ kcal/mol in the presence of ssDNA) showed that the mutation would decrease the interaction between PprI and DdrO, providing explanations for the FRET assay results (Fig. 4a, e). Moreover, DdrO mutant proteins bearing alanine substitutions of key CSR residues (E116A, L117A, R118A, and G119A) were purified and subjected to PprI cleavage and ssDNA activation assays to further confirm possible interactions between the CSR loop and ssDNA (Fig. 4f). In the absence of ssDNA, all the DdrO mutants except L117A exhibited considerably decreased cleavage by

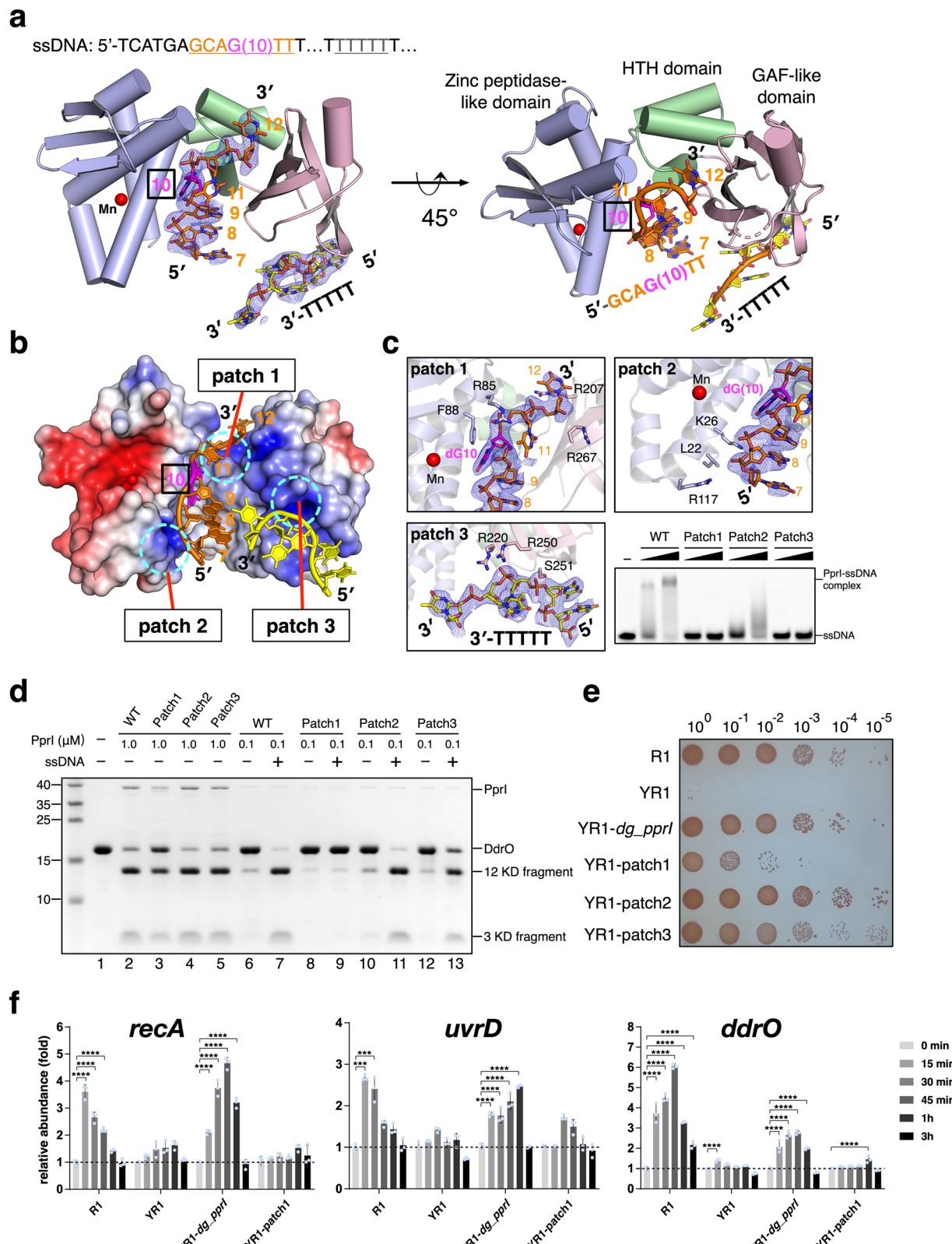

PprI (1 μM, lanes 1-10), suggesting that these residues are involved in the interaction. Such weakened cleavage of E116A and G119A was restored with the addition of ssDNA (lanes 14 and 20). In contrast, R118A mutation led to unrecoverable cleavage in the presence of ssDNA (lane 18), indicating that the Arg118-ssDNA interaction was required for PprI cleavage in the presence of ssDNA (Fig. 4b). Together, these results provided insight into a plausible activation mechanism of

PprI, that ssDNA enhances the PprI-DdrO interaction by restraining or locating the CSR in the cleave site (Fig. 4b).

## Dynamic monomer-dimer equilibrium of PprI involved in DNA damage response

PprI in complex with ssDNA crystallized in space group $P3_121$, with a dimer molecule in the crystallographic asymmetric unit. The PprI

**Fig. 3 | Interactions between PprI and ssDNA. a** Overall structure of PprI-ssDNA complex. DG-PprI and ssDNA are labeled and shown as cartoon and sticks, respectively. A schematic of the DNA substrate used for crystallization is shown on top with colors corresponding to those observed in the PprI-ssDNA structure below. The electron density of two segments of ssDNA (5'-GCAGTT and 3'-TTTTT) is shown in blue with the refined 2Fo-Fc contoured at 1σ. The catalytic metal ion is shown as sphere and colored red. **b** The ssDNA binding patches (patches 1-3) of DG-PprI are labeled and shown with electrostatic surface potentials. Blue and red represent the positive and negative charge potential at the + and -5kT e⁻¹ scale, respectively. **c** Close view of three ssDNA binding patches of DG-PprI. The EMSA at the bottom-right corner showing the abolished or decreased ssDNA binding of DG-PprI triple mutants (patch 1-3 mutants). The reaction conditions is the same as in Fig. 1c. **d** Cleavage and ssDNA activation assays of the DG-PprI triple mutants (patch 1-3 mutants). The cleavage assays in the absensece of ssDNA were performed using

1 μM of DG-PprI (lanes 2-5). For ssDNA activation assays (lanes 6-13), 0.1 μM of DG-PprI was used under the same reaction conditions as in Fig. 2a. **e** Phenotypic analyses of the DG-PprI triple-mutant comlementary strains (patch 1-3 mutants). Wild-type strain (R1), *dr_pprI* knockout strain (YR1), and *dg_pprI* complementary strains (YR1-*dg_pprI* for the wild-type DG-PprI and YR1-patch 1-3 for DG-PprI triple mutants) were spotted on TGY medium following 4 kGy gamma radiation treatments. **f** Quantitative real-time PCR analysis of the gene expression levels of *recA*, *uvrD*, and *ddrO*. Total RNA was isolated from the R1, YR1, YR1-*dg_pprI* and YR1-patch1 mutant strains under normal growth conditions and after 8 kGy gamma radiation treatments at different time points during the recovery (15 min, 30 min, 45 min, 1 h and 3 h). Data represent the means of the three replicates, and the bars represent their standard deviations. One-way ANOVA method followed by Tukey's post-hoc test was performed to compare the significant differences: ***$p < 0.001$ and ****$p < 0.0001$. Source data are provided as a Source Data file.

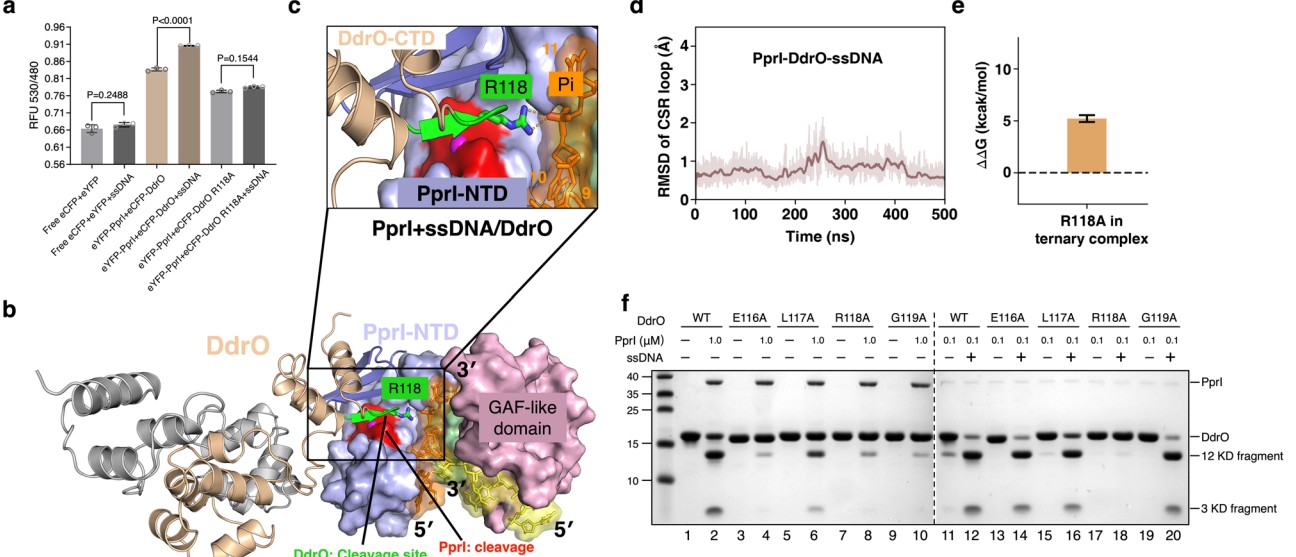

**Fig. 4 | ssDNA enhances the PprI-DdrO interactions. a** FRET showing the interactions between PprI and DdrO. The transfer efficiency is represented by the RFU ratio of 530/480 nm. Data represent the means of the three replicates and the bars represent their standard deviations. One-way ANOVA method followed by Tukey's post-hoc test was performed to compare the significant differences. **b** A Snapshot of all-atom molecular dynamics (MD) simulations of the PprI-DdrO-ssDNA model. The DG-PprI, DdrO, and ssDNA are labeled and shown as surface, cartoon and sticks, respectively. The cleavage center of DG-PprI and the CSR loop of DdrO (Arg118) are highlighted in red and green, respectively. **c** Close view of cleavage center of PprI and CSR of DdrO during MD simulations. R118 of DdrO interacts with the phosphate group of ssDNA through hydrogen bonds. **d** RMSD of CSR loop in the ternary complex model during 500 ns simulations. **e** The relative binding free energy change (ΔΔG) of R118A mutation in the presence of ssDNA was estimated through free energy perturbation (FEP) calculations. Data are presented as mean values +/− SEM from 3 independent experiments. **f** Cleavage (lanes 1–10) and ssDNA activation (lanes 11-20) assays of the DdrO mutants (E116A, L117A, R118A, and G119A). The reaction conditions is the same as in Fig. 3d. Source data are provided as a Source Data file.

homodimer in the PprI-ssDNA structure exhibits an extended overall structure in a face-to-face fashion, with two DG-PprI protomers interacting with each other through the N-terminal β-strands (Fig. 5a), which is different from the side-by-side dimer configuration of DG-PprI and DD-PprI apo structure solved previously[33] (Fig. 5b). PISA server[50] was used to calculate the solvent-accessible surface area of these two dimer interfaces. Compared with the side-by-side DD-PprI dimer containing 533.8Å² interface area, the two molecules buried approximately 840.6Å² of surface in the face-to-face DG-PprI dimer, which was contributed by salt bridges and hydrogen bonds. Notably, a β-pin motif (residues 69-73, the β2-β3 loop) from one promoter protrudes into the cleavage site of a neighboring protomer, with His72 interacting with the catalytic metal ion of the neighboring DG-PprI protomer (Fig. 5a). Thus, in addition to the ssDNA and conserved HEXXH motif supporting the catalytic metal ion binding, the face-to-face dimer formation may also contribute to the cleavage site formation of DG-PprI. We further verified the dimer formation of DG-PprI in solution using size exclusion chromatography. In addition to the dominant peak

corresponding to monomeric DG-PprI, a small portion of DG-PprI dimer was observed (Fig. 5c), which was also confirmed by formaldehyde crosslink assay (Supplementary Fig. 10a). However, the addition of ssDNA did not alter the proportion of dimeric DG-PprI in solution or the intermolecular interactions between PprI proteins (Supplementary Fig. 10b, c), indicating that ssDNA is not involved in the PprI monomer-dimer equilibrium in solution.

Given the varied dimer interfaces observed in the PprI-ssDNA (face-to-face) and apo-PprI (side-by-side) structures, three types of dimer interface mutants bearing alanine substitutions were purified to further confirm the dimer formation of DG-PprI: a quadruple-mutant (D69A/E71A/H72A/R73A) at the dimer interface of the PprI-ssDNA structure (face-to-face interactions), a double-mutant (equivalent H46A/F58A) at the interface of the DG-PprI structure (side-by-side interactions), and the combined hexa-mutant (H46A/F58A/D69A/E71A/H72A/R73A). These mutant proteins were first examined by circular dichroism spectra showing their proper folding (Supplementary Fig. 10d). Interestingly, neither the quadruple-mutant (4mut) nor the

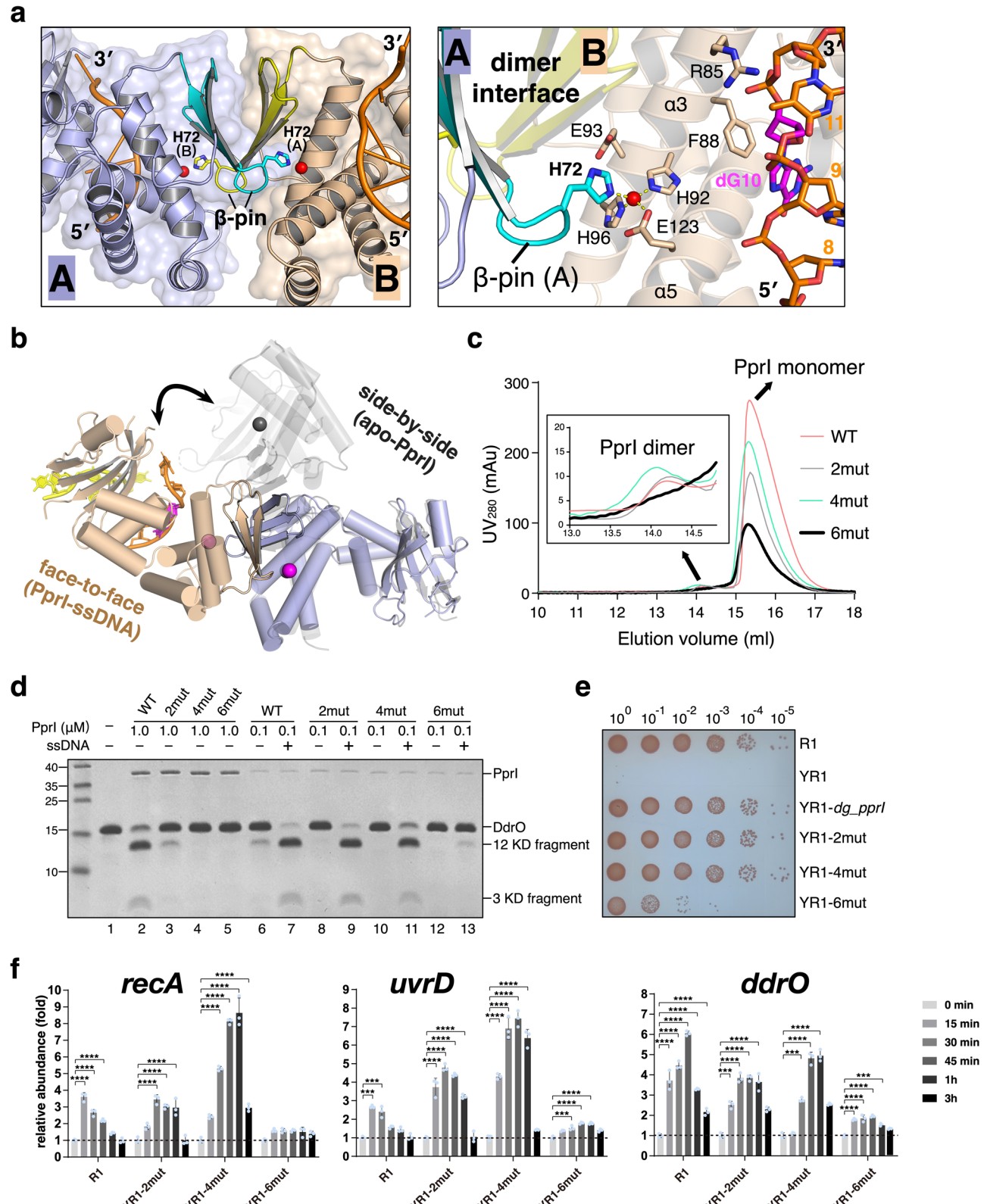

double-mutant (2mut) exhibited an altered proportion of dimeric DG-PprI in solution (Fig. 5c). In contrast, the hexa-mutant (6mut) destroying both interfaces showed an imperceptible peak corresponding to the DG-PprI dimer (Fig. 5c) as well as significant decreased FRET signals (Supplementary Fig. 10e), indicating the coexistence of both interfaces and dynamic behavior of the monomer-dimer equilibrium of DG-PprI in solution. We next checked whether these

mutations had effect on their ssDNA activated cleavage activity (Fig. 5d and Supplementary Fig. 7). Compared with the wild-type protein, the double-mutant exhibited severely impaired protease activity in the absence of ssDNA (lane 3), and cleavage activity was not detected for the quadruple- and hexa-mutant proteins (lanes 4 and 5), indicating that the face-to-face and side-by-side dimer interfaces of DG-PprI were important for its inherent cleavage. While the addition of ssDNA

**Fig. 5 | Dynamic monomer-dimer equilibrium of PprI. a** The left panel shows DG-PprI dimer in a face-to-face fashion in PprI-ssDNA structure. Two DG-PprI protomers are labeled and colored in distinct colors (protomer A in slate and cyan; protomer B in wheat and yellow). The His72 residue of β-pin is labeled and shown as stick. The right panel shows the close view of His72 interactions at the face-to-face dimer interface. **b** Superposition of two PprI dimer configurations with distinct colors (face-to-face dimer in slate and wheat; side-by-side dimer in black). **c** Size exclusion chromatography of wild-type (WT) DG-PprI and dimer-interface mutants (2mut: double-mutant at side-by-side interface, 4mut: quadruple-mutant at face-to-face interface, and 6mut: combined double- and quadruple-mutant) on Superdex 200 10/300 GL column. The peaks correspond to monomeric or dimeric PprI

proteins are labeled. **d** Cleavage (lanes 2–5) and ssDNA activation (lanes 6-13) assays of the dimer interface mutants (2mut, 4mut, and 6mut). The reaction conditions is the same as in Fig. 3d. **e** Phenotypic analyses of the DG-PprI comlementary strains (dimer interface mutants) following 4 kGy gamma radiation treatments. **f** Quantitative real-time PCR analysis of the gene expression levels of *recA*, *uvrD*, and *ddrO* were performed as in Fig. 3f. Data represent the means of the three replicates, and the bars represent their standard deviations. One-way ANOVA method followed by Tukey's post-hoc test was performed to compare the significant differences: $***p < 0.001$ and $****p < 0.0001$. Source data are provided as a Source Data file.

significantly enhanced DdrO cleavage by the double- and quadruple-mutants (lanes 9 and 11), it failed to stimulate the protease activity of hexa-mutant DG-PprI (lane 13). Combined with their oligomeric status in solution, these results suggested the necessity of dimer equilibrium (face-to-face or side-by-side) of PprI for its ssDNA activation. The biochemical observations were also supported by our in vivo data (Fig. 5e). While overexpression of the double- and quadruple-mutants fully compensated the phenotype of the YR1 strain, the hexa-mutant complementary strain (YR1-6mut) was extremely sensitive to gamma radiation, which was consistent with the transcriptional patterns of DDR genes in vivo (Fig. 5f): in contrast to 5–10-fold inductions of *recA*, *uvrD*, and *ddrO* in the double-mutant and quadruple-mutant complementary strains, transcriptional levels of these genes were maintained at basal level (less than 2-fold increase) after gamma radiation treatments.

## Discussion

*Deinococcus* species, including *D. radiodurans* and *D. geothermalis*, have been shown to tolerate high doses of ionizing radiation due to their super-effective DNA repair system. Research over the past two decades has continued to highlight the features of the PprI-DdrO system using biochemical, structural and genetic methods. However, given the constant expression level of PprI after DNA damage, questions remained regarding the activation mechanisms of PprI cleavage.

In the present study, we found that DG-PprI can directly bind ssDNA but not dsDNA or RNA (Figs. 1c and 3a). Aided by biochemical, structural, phenotypical, and transcriptional analyses, we showed that ssDNA significantly activated the protease activity of DG-PprI both in vivo and in vitro. Thus, ssDNA per se could serve as the primary signal for DNA damage sensing in *Deinococcus* species. Although ssDNA is required for both DdrO and LexA digestion, the strategy employed by the PprI-DdrO system is distinct from the *E. coli* SOS response in the following aspects: (1) in contrast to the ssDNA-binding properties of *E. coli* homologue, RecA proteins in *Deinococcus* preferred to bind dsDNA over ssDNA[51], which may not form sufficient RecA-ssDNA filaments for LexA autocleavage. (2) Compared to radiosensitive bacteria such as *E. coli*, *Deinococcus* species have much higher intracellular Mn concentrations (ranging from 0.2 to 4 mM, Supplementary Fig. 11a) as well as the manganese-to-iron ratio (0.24 in *D. radiodurans* and 0.46 in *D. geothermalis*)[52,53], which contribute to the proteome protection critical for not only antioxidation but also DNA repair. Indeed, despite low concentrations of Zn or Mn ions (20 μM) being able to activate the DG-PprI digestion in vitro, Mn was effective at high concentrations (Supplementary Fig. 11b, c, d). In addition to PprI cleavage, manganese was frequently observed in structures of *Deinococcus* proteins and served as the cofactor for enzymatic activities[54]. (3) consensus sequences of SOS box varied in different organisms, resulting in two phases of SOS induction depending on the extent of DNA damage. However, the RDRM motif of DdrO binding has been found and conserved only in *Deinococcus* species to date. Moreover, while the depletion of LexA in *E. coli* resulting in growth delay and an SOS response phenotype, DdrO is an essential protein for

cell viability[32], indicating that the PprI-DdrO system plays a vital role in fundamental processes in vivo beyond DNA damage response.

Despite the varied protein structures and enzymatic activities of DNA repair proteins, the general processes of DNA repair in bacteria and eukaryotes are shared in terms of DNA damage response, lesion removal, and gap filling. The ssDNA was readily found at DNA damage sites after DNA end resection processes, which were initiatively generated by coordinated actions of nucleases and helicases, *e.g.*, Mre11-Rad50-Nbs1/Xrs2-CtiP and Dna2-Sgs1 in eukaryotes[55,56], RecBC/AddAB and RecJQ/UvrD in bacteria[57,58], and the HerA-NurA complex in archaea[59]. Moreover, it has been shown that the newly generated ssDNA by DNA end resection is also involved in the regulation of DSB repair choice between homologous recombination and non-homologous end joining in eukaryotes[55]. It is worth noting that the well-documented RecBC end resection complex is absent in *D. radiodurans*, while the bacterial type RecJQ/UvrD and archaeal type HerA-NurA operon coexist in cells, with physical interactions between RecJ and NurA nucleases[60,61]. Given the ESDSA repair pathway proposed in *D. radiodurans* requiring extensive synthesis of ssDNA (20-30 kb) for recombination[62], ssDNA engendered by DNA end resection appears to play a pivotal role in PprI-DdrO mediated DDR except recombinational repair processes in *Deinococcus* species.

ssDNA activated DG-PprI cleavage in a length-dependent manner, which required dimerization of PprI. In the PprI-ssDNA structure, the patch 1 interface containing two GAF-like domain residues (Arg207 and Arg267) was critical for ssDNA binding and activation, with ssDNA anchored to patch 3 interface (Fig. 3a, b); this was consistent with the appropriate length of ssDNA characterized in our biochemical assays (Fig. 1d). In some proteins, the GAF-like domains bind cyclic nucleotides such as cAMP and cGMP and function as receptors for signal transduction[63,64]. Thus, although the GAF-like domain of PprI appeared to not be essential for its inherent cleavage in *E. coli* or in vitro[39], other ligands may regulate the PprI-DdrO system by modulating the ssDNA binding and activation through GAF-like domain interactions.

In the absence of ssDNA, PprI exhibited limited protease activity in vitro and adopted an open cleavage site with side-by-side dimer formation (Fig. 5b, d). However, a distinctive face-to-face dimer interface was observed in our PprI-ssDNA structure (Fig. 5a, b). A β-pin motif from a neighboring PprI protomer protrudes into the cleavage site, together with the HEXXH motif, coordinating the catalytic metal ion. Given the configuration of the cleavage site and nearby ssDNA observed in the PprI-ssDNA structure, ssDNA could modulate cleavage site formation of PprI, further stabilizing or locating the CSR loop of DdrO (Fig. 4b), which resembled the RecA filaments mediated LexA autocleavage[65]. Interestingly, a bacterial CBASS immune pathway in response to DNA damage was recently reported, which is mediated by CapP (protease) and CapH (transcription factor)[38]. Despite the low protein sequence identity, the overall structure of CapP could be aligned with DG-PprI structure (Supplementary Fig. 2b). Moreover, ssDNA also activated the CapH cleavage by CapP, leading to the derepression of downstream transcription. Thus, in addition to the SOS response, derepression systems mediated by ssDNA triggered proteolytic cleavage of transcription factors appear universal and

effective for the stress response in bacteria. The results reported here potentially facilitate the development of drugs tackling antimicrobial resistance.

## Methods

### Strains and Culture

*D. radiodurans* R1 (ATCC 13939, R1), *D. geothermalis* (DSM 11300) and their derivatives were grown in TGY broth (0.5% tryptone, 0.3% yeast extract, 0.1% glucose) or on TGY plates with 1.5% (w/v) agar powder at 30 °C or 45 °C, respectively. *E. coli* trans5α and BL21 (DE3) strains were cultivated in LB broth (1% tryptone, 0.5% yeast extract, 1% NaCl) or on LB plates with 1.5% (w/v) agar at 37 °C. Kanamycin (40 μg/ml for *E. coli*, 8 μg/ml for *D. radiodurans*), ampicillin (100 μg/ml for *E. coli*), and chloramphenicol (4 μg/ml for *D. radiodurans*) were used for antibiotic selection. All the strains and plasmids are listed in Supplementary Table 2.

### Cloning and strain constructions

DG-PprI, full-length (residues 1-289) and GAF-like domain-truncated DG-PprI (residues 1-176) were constructed as previously described[25]. Briefly, the fragments were amplified by PCR and cloned into pET28a-HMT expression vector using NdeI and BamHI restriction enzyme sites. Amplification and site-directed mutagenesis were carried out using PrimeSTAR HS DNA polymerase (Takara). For site-directed mutagenesis, the amplified product was treated with DpnI and transformed into DH5α following temperature cycling. All pET28a vectors were sequenced before transforming into the *E. coli* BL21(DE3) for protein expression. To obtain complementary strains for phenotypic assays, PCR amplified wild-type, truncated, and site-mutant *dg_pprI* were cloned into the continuous protein expression shuttle plasmid pRADK containing strong promoter groEL[66,67] and transformed into the *dr_pprI* knock-out strain (YR1). All the primers and DNA substrates are listed in Supplementary Table 3.

### Protein expression and purification

DG-DdrO, DG-PprI and their derivatives were expressed and purified as previously described[25]. Briefly, the expression strains were grown in LB broth at 37 °C to an optical density at 600 nm of 0.6-0.8, followed by adding isopropyl-β-D-thioga-lactopyranoside (IPTG) at a final concentration of 0.2 mM. The harvested cell pellets were lysed by sonication and centrifuged at 14,000 rpm for 30 min at 4 °C. The supernatant was purified by an AKTA Purifier system with HisTrap HP column and Heparin HP column, followed by gel filtration chromatography analysis. Tag-removed proteins were concentrated, aliquoted in gel filtration buffer: 150 mM NaCl, 20 mM Tris-HCl (pH 8.0), and stored at −80 °C after flash frozen.

### Crystallization and structure determination

DG-PprI was concentrated to ~10 mg/ml and the apo crystals were grown using the drop vapor diffusion method at 289 K over wells containing 1.7 M $(NH_4)_2SO_4$, 0.1 M Bis-tris (pH 6.5), and 2.5 mM $MnCl_2$. The freshly prepared DG-PprI protein (~8 mg/ml) and 29nt ssDNA were mixed at a 1:2 molar ratio for PprI-ssDNA crystallization. The complex crystals were grown in 0.3 M potassium sodium tartrate tetrahydrate, 20% polyethylene glycol 3350, and 2.5 mM $MnCl_2$. Cryocooling was achieved by stepwise soaking with crystals in reservoir solution containing 10, 20, and 30% (w/v) glycerol for 5 min, followed by flash freezing in liquid nitrogen. The diffraction intensities were recorded on beamline BL17U1 at Shanghai Synchrotron Radiation Facility (Shanghai, China) and were integrated and scaled with the XDS suite. The structure was determined by molecular replacement using DD-PprI (PDB ID: 3DTI) as the search model[33]. Structures were refined using PHENIX[68] and interspersed with manual model building using COOT[69]. All the residues are in the most favorable and allowed regions of the Ramachandran plot. The final structures contain residues 21–188

and 203–275. The omit map for sulfate from apo structure of the DG-PprI and the omit map for manganese ion and ssDNA from PprI-ssDNA complex are shown in Supplementary Fig. 12. The statistics for data collection and refinement are listed in Supplementary Table 1. All structural figures were rendered in PyMOL (www.pymol.org). The ssDNA used in this crystalization is listed in Supplementary Table 3.

### Prediction of PprI-DdrO-ssDNA structure by AlphaFold2

The amino acid sequences of DG-PprI and DG-DdrO were uploaded to the AlphaFold2 prediction server[40,41] to obtain PprI-DdrO (full-length monomeric DdrO) complex model. The top-ranked complex model (Supplementary Table 4, Supplementary Fig. 9) was selected and aligned with the DdrO dimer structure (Protein Data Bank [PDB] ID: 6JQ1)[25] solved in the previous study using its C-terminal domain to obtain the PprI-DdrO (full-length dimeric DdrO) complex model. The binary model was then overlayed on the PprI-ssDNA structure to obtain a ternary model of PprI-DdrO-ssDNA.

### Circular Dichroism (CD) spectra

After phosphate buffer (PB) exchange, purified DG-PprI proteins were concentrated to 0.2 mg/ml. Circular dichroism spectra were measured on a J-1500-150ST spectrometer (JASCO, Japan). PB buffer was used as the baseline. The CD spectra of DG-PprI proteins were recorded as average of three scans in the wavelength range 190-260 nm.

### Formaldehyde crosslink assay

DG-PprI proteins were concentrated to 1.7 mg/ml in 50 mM HEPES (pH 8.0) buffer containing 0.1 mM EDTA. The crosslink assay was performed in reaction buffer containing a final concentration of 25 mM formaldehyde, followed by incubation at 30 °C for 30 min. The crosslinking results were detected by SDS-PAGE electrophoresis.

### Electrophoretic mobility shift assay (EMSA)

EMSA reaction mixture consisted of 150 mM NaCl, 20 mM Tris-HCl (pH 8.0), 0.1 mg/ml Albumin from bovine serum (BSA), 0.1 μM 5′-FAM-labeled ssDNA, and DG-PprI proteins (0, 1 and, 2 μM), and were incubated at room temperature for 30 min. Samples were separated on 10% native polyacrylamide gels in 1xTB buffer. Gels were imaged in fluorescence mode on Typhoon FLA 9500 (GE, USA). The substrates used in this experiment are listed in Supplementary Table 3.

### PprI cleavage and ssDNA activation assays

The cleavage assays with high concentration of DG-PprI (1 μM) were performed in the absence of ssDNA as previously described[25]. Briefly, a final concentration of 8 μM DG-DdrO and 1 μM DG-PprI were incubated at 37 °C for 30 min in the conditions of 200 mM NaCl, 20 mM Tris−HCl 8.0, 1 mM DTT and 2 mM $MnCl_2$. The activation assays were carried out under the same reaction conditions containing ssDNA (0.1 μM) and low concentration of DG-PprI (0.1 μM) at 37 °C for 30 minutes. The cleavage results were detected by Tricine-SDS-PAGE. The substrates used in this experiment are listed in Supplementary Table 3.

### Phenotypic assays

Cells were grown to the early exponential phase ($OD_{600}$ = 0.6−0.8) without antibiotics and resuspended in sterile PBS buffer. After gamma radiation (4 kGy) treatments, cells were diluted and dotted onto TGY plates. Plates were cultured at 30 °C for 2-3 days.

### Fluorescence resonance energy transfer (FRET) measurements

PprI and DdrO proteins fused with N-terminal eYFP or eCFP, respectively, were mixed at a 1:1 molar ratio (5 μM each) within a black bottom opaque 384-well plate (Corning, USA) in reaction buffer: 200 mM NaCl, 20 mM Tris (pH 8.0), 20 mM $MnCl_2$. FRET measurements were performed using SpectraMax M5 plate reader (Molecular Devices, USA) with excitation set at 440 nm, and emission measured from

460–600 nm. The transfer efficiency is represented by the RFU ratio of 530/480 nm. All reactions were independently repeated at least three times. The substrates used in this experiment are listed in Supplementary Table 3.

### Quantitative real-time PCR (qRT-PCR)

Wild-type (R1), YR1, and complementary strains were cultured in TGY broth until the $OD_{600}$ reached 1.0, subsequently treated with γ-ray (8 kGy) for 4 h. After irradiation, cells were harvested at different time points during the recovery (15 min, 30 min, 45 min, 60 min and 3 h). Total RNA extraction and reverse transcription PCR were performed as previously described[67]. Briefly, total RNA was extracted with TransZol Up Plus RNA Kit (TransGen Biotech) when cells were grown at an absorbance of 1.0 at 600 nm. Complementary DNA was synthetized from 1000 ng of RNA using the PrimeScript Reverse Transcriptase (TAKARA). SYBR Premix Ex Taq (Takara, Japan) was used to perform quantitative real-time PCR, and the housekeeping gene *dr_1343* was used for normalization[70]. All reactions were independently repeated at least three times. The primers used in this experiment are listed in Supplementary Table 3.

### Statistical analysis and reproducibility

All the experiment results have been successfully repeated for at least three times. Results of qRT-PCR and FRET measurements assays were determined using GraphPad Prism 8. The results represented the means and standard deviations (SD) of three independent experiments. One-way ANOVA method followed by Tukey's post-hoc test was performed to compare the significant differences.

### Molecular dynamics simulations

All MD simulation systems are listed in Supplementary Table 5. The size of solution system for ternary model was $14 \times 10 \times 8$ $nm^3$. All systems were solvated with TIP3P water[71]. $K^+$ and $Cl^-$ ions were added to the bulk water at a salt (KCl) concentration of 150 mM. $Mn^{2+}$ was added with reference to the position in the structure of PprI-ssDNA solved in this study. The final system sizes are approximately 100,000 atoms. After an initial equilibration, all systems were run using GROMACS 2020.6[72] for 575 ns in triplicate for better statistics. The AMBER ff99SB-ILDN force field[73] for proteins and the parmbsc1 force field[74] for DNA were used. The CM 12-6 parameters[75] were used for $Mn^{2+}$ ions. The temperature was maintained at 310 K and the pressure at 1 atm using V-rescale[76] and Parrinello-Rahman[77], respectively. The cutoff distance of Van der Waals interactions was 12 Å. The long-range electrostatic interactions were treated using the particle mesh Ewald method[78]. The covalent bonds with hydrogen atoms were constrained by the LINCS algorithm[79], which allows a time step of 2 fs. All trajectory analyses were performed by VMD[80] for the last 500 ns simulation. The RMSD of the CSR loop was measured after aligning the zinc peptidase-like domain of PrpI for comparisons.

### The estimation of differences in binding affinity due to in silico mutagenesis

The relative binding free energy differences (ΔΔGs) of alanine substitutions were calculated by Free Energy Perturbation[47–49], and combined with Hamiltonian Replica-Exchange Molecular Dynamics[81] for efficient convergence. The Free energy changes for alanine substitutions were estimated in both the bound state (PprI-DdrO-ssDNA for ternary model) $\Delta G_{bound}$ and the free state (isolated DdrO for the ternary model) $\Delta G_{free}$ using Gromacs 2020.6. Thus, the binding free energy change caused by residue mutation is estimated as $\Delta\Delta G = \Delta G_{bound} - \Delta G_{free}$. PMX[82,83] was used for preparing the dual-topology files with amber99sb-ildn force field. For each mutation, 24 windows of sequential annihilation of electrostatics and van der Waals were set up, and started from the same equilibrated system. Every alanine substitutions calculation was performed for at least 62 ns (1.3 ns × 24 windows × 2 states). The soft-core potentials[84] (α, the power for lambda term, the power of the radial term, and sigma were set to 0.5, 1, 6, and 0.3, respectively) were used during simulations. The exchange between neighboring windows are attempted every 1 ps. Hamiltonians of the systems were saved every 0.2 ps. ΔΔGs and their statistical errors were estimated from the last 1 ns simulation of each window using the Multistate Bennett Acceptance Ratio (MBAR) method[85] in Alchemical Analysis[86].

### Reporting summary

Further information on research design is available in the Nature Portfolio Reporting Summary linked to this article.

## Data availability

The raw data of cleavage assays, activation assays, EMSA experiments, phenotype assays, qRT-PCR, FRET assays, CD spectra and FEP calculations are provided in a Source Data file. The coordinates and structure factors have been deposited to Protein Data Bank with accession codes 8SLM (PprI-apo) and 8SLN (PprI-ssDNA complex), which are also provided as Supplementary Data 1. The PDB database used in the study includes PDB IDs: 3DTI and 6JQ1. The input set of coordinates for molecular dynamics simulations is provided as Supplementary Data 2 and the output set of coordinates for molecular dynamics simulations is provided as Supplementary Data 3. Source Data are included in the Source Data file. Source data are provided with this paper.

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

## Acknowledgements

We would like to thank the staff at the Shanghai Synchrotron Radiation Facility (SSRF in China) for assistance in data collection. This work was supported by the National Natural Science Foundation of China (32370028 [H], 32222001 [Zhao], 32200016 [L], and U1967217 [Zhou]), the National Key Research and Development Program of China (2021YFD1400500 [Zhao], 2021YFA1201201 [Zhou] and 2021YFF1200404 [Zhou]), and Zhejiang Provincial Natural Science Foundation of China (LDQ23C050002 [Zhao], LQ23C010002 [L]).

## Author contributions

Y.H. and Y.Z. conceived the study. H.L. and Y.Z. designed the experiments. R.Z. and T.X. designed the molecular modeling. H.L. and Z.C performed protein purification, crystallization, phenotypic assays, acquisition of data and biochemical assays. Y.Z. determined and analyzed the structure. Z.C. and S.Z. performed the FRET assays. T.X. performed the binding complex model construction, molecular dynamics simulations, and free energy calculations. S.Suo and S.Song performed molecular cloning and constructed the mutants. L.W., H.X. and B.T. took part in mutant strain construction and data analysis. H.L., Y.Z., T.X., R.Z. and Y.H. wrote the manuscript. All authors took part in data analysis.

## Competing interests

The authors declare no competing interests.
