## [Peer Review File NEW · Nature Communications]

The Deinococcus protease PprI senses DNA damage by directly interacting with single-stranded DNAReviewer #1 (Remarks to the Author):

The authors of this manuscript aim to elucidate the mechanism of activation of the protease PprI cleaving DdrO, the transcriptional repressor targeted by this protein in the radioresistant bacterium *Deinococcus geothermalis*. These two proteins play a major role in the expression of DNA repair genes in several *Deinococcaceae* after exposure to irradiation.

This manuscript contains numerous experiments and several interesting results, in particular the co-crystallization of PprI with single-stranded DNA, but much information is missing, making it impossible to clearly explain how certain experiments were carried out, and therefore the relevance of several results. Several previous works published on this subject, sometimes performed by other research teams, are not cited and/or discussed according to the results of this manuscript.

-The authors justify adding MnCl₂ for protein crystallization, as this ion would be preferentially used for DdrO cleavage. However, the manuscript cited as a reference did not aim to compare different ions for the cleavage efficiency of PprI. In contrast, another manuscript, that is not cited here (Blanchard et al, 2017) highlighted that Zn ions were preferred for the protease activity of PprI from *Deinococcus deserti*. This result is in agreement with the crystallographic data of the *D. deserti* apo protein PprI published in 2009 (Vujicic-Zaggar et al. 2009): PDB 3TDE (inactive apo-enzyme), 3DTI (1 Zn, active enzyme) and 3DTK (2 Zn at the active site, inactive enzyme because too many divalent cations inactivate the enzyme (problem of accessibility of the water molecule as for the thermolysin). The catalytic metal is coordinated by the same HEISH motifs in both PprI proteins (Figure S1) and the two structures are superimposable. The authors do not discuss this important difference even though these two proteins are very similar in sequence and structure. Furthermore, in Blanchard et al, 2017, the authors also showed that metal concentration plays an important role for cleavage efficiency of IrrE and the activation occurred *in vitro* with Zn²⁺, Mn²⁺, Fe²⁺, CO₂⁺ but not by Ni²⁺, Ca²⁺, Mg²⁺. They also show that *in vivo* only Zn is capable of activating IrrE. The high concentration used here (2 mM) could have an inhibitory role and explain why no cleavage is observed with Zn (Figure 2C). This is an important point since of all the experiments carried out with the wild-type protein and the numerous mutants were tested with this high concentration of Manganese and not the Zn ion.

- Figure 1C: Several authors of the manuscript already published EMSA assays showing that PprI is able to bind to double-stranded DNA containing an RDR motif (Lu et al., DNA repair 2011). How can they explain the differences between IrrE from *D. deserti*, *D. radiodurans* and *D. geothermalis* with these apo-protein structures data and also thanks to the structure obtained with single-stranded DNA? What are the structural bases for these differences? How do the authors settle these discrepancies?

- Figure 1E No shift is observed in the presence of a short oligonucleotide (10 nt), whereas a stimulation of cleavage is observed in the presence of an oligonucleotide of the same size (Figure 2D): How do the authors settle these two experiments?

- No information is given about the conditions of expression of the wild-type PprI protein, and the different mutated forms of this protein in *D. geothermalis* from the shuttle vector. Did they clone the genes under the control of the PprI promoter or of another promoter? In particular, the authors indicate an overexpression of these proteins *in vivo* (Lines 156, 193, 307). Is protein expression from the vector continuous or induced and in the latter case how? Furthermore, the expression kinetics of *ddrO* regulon genes (*recA*, *uvrD*, *ddrO*) under PprI control in the YR1-dg_pprI strain are not comparable with those in wt cells (Figures 3F and 5F). They also differ *in vivo* (Figure S4C) and between experiments for the same strain (Figure 3E and Figure S4C). The wild-type protein is less effective than the mutants tested (Figure 5E). These differences are not explained by the authors.

- The authors did not explain the choice of the housekeeping gene *dr_1343* for qPCR normalization

- Figure 4b: little information is given on the parameters used to build the model: The manuscript presents predicted 3D structures of proteins/complexes that were obtained with AlphaFold2, more details regarding the parameters used for these predictions and an assessment of the quality/confidence levels of these predictions should be given and discussed. This is critical for the

comparisons for instance where side chains are compared. What is the confidence in the predicted structures for these regions, and consequently would they impact the interpretation? Could they provide a model with the full DdrO protein and not just with CTD?

- Two different organizations of the PprI protein were observed in the crystals obtained with the apo-protein and the protein bound to the single-stranded oligonucleotide (side by side and face to face respectively), but only a very small number of protein dimers were observed in solution even after attempting to crosslink the monomers. No change was observed when ssDNA was added (Figure 5 and Figure S7A and b). The mutations introduced at specific locations (Patch 1 and 2) did not decrease the cleavage of ddrO, excepting if 6 different mutations were introduced at specific positions which would impair the equilibrium monomer/dimer of PprI. What conclusion can be drawn from these results? What are the consequence of a PprI dimerization on the model of the interaction between DdrO and PprI (Figure 4b)?.

- The model proposed by the authors about the role of single strand DNA in vivo may be questionable. The authors repeatedly showed that single-stranded DNA is not required for complete cleavage of DdrO by PprI when the cleavage assay was performed at a respective concentration of the two partners of 8/1. Stimulation is observed when the ratio is 8/0.1. In vivo, the PprI protein is reputed to be constitutively expressed, whereas the DdrO expression is repressed by the protein itself. After irradiation, DdrO is cleaved by PprI, but continued expression of the DdrO protein leads cells to an apoptotic pathway (Devigne et al, 2015). Consequently, why ssDNA would be used to activate PprI if the concentration of PprI into the cells remains higher than that of DdrO in vivo?

Reviewer #2 (Remarks to the Author):

In this study, Lu et al. shed light on the mechanisms underlying the response of the radiation resistant *Deinococcus* bacteria to DNA damage. In these bacteria, the classical SOS response system involving notably LexA and RecA is missing and is instead replaced by a metalloprotease, PprI/IrrE, that cleaves a transcriptional repressor, DdrO, in response to DNA damage. Several articles in the recent years have unveiled the molecular details underlying this process, but so far little is known regarding the mechanisms responsible for activation of PprI/IrrE. Indeed, unlike LexA, PprI/IrrE is constitutively expressed and thus needs to be activated to perform its cleavage of DdrO. Here, the authors report the crystal structure of apo-PprI from *D. geothermalis*, which is very similar to the previously reported structure of *D. deserti* PprI, but more importantly, they reveal that ssDNA can strongly activate the protease activity of PprI and report the crystal structure of ssDNA-bound PprI. This structural and biochemical study, supported by in vivo mutagenesis analysis, represents an important advance in our understanding of the DNA damage response in *Deinococcus* bacteria, identifying ssDNA, which is a by-product of large-scale DNA damage, as a key activator of the response pathway. Overall, the manuscript is very well written and clear, and the presented experiments have been well performed and support the conclusions.

A few points listed below should nonetheless be addressed before this manuscript can be published.

Major points:

- i) The quality of the structures, and in particular the geometry, could certainly be improved to reduce the clashscore and sidechain outliers (and lower the Rfree). For the PprI-ssDNA structure, a more detailed description of the visible DNA bases and their fit to the density and contacts with the protein should be provided – could the authors clearly define the polarity of the two fragments?
- ii) ssDNA used for EMSAs and protease assays should ideally be polydT to avoid the formation of secondary structure, hairpins or partial duplexes. Authors should comment on this and provide evidence that their ssDNA fragments were indeed fully single-stranded.
- iii) Timecourse experiments to follow the kinetics of the protease activity of PprI would be more insightful than single time point experiments, especially for comparing mutants.
- iv) The second ssDNA fragment bound to patch 3 of PprI is actually located at the interface with a neighboring symmetry-related molecule in the crystal – could this be a crystal packing artefact?

This should perhaps be discussed in view that mutations in this patch had little effects on the protease activation.

v) Authors state that they used AlphaFold2 to model the ternary structure of PprI-DdrO-ssDNA, but this is incorrect. They used AF2 to model PprI binding to the C-terminal region of DdrO and then overlaid this model on the PprI-ssDNA structure to build a ternary model. This should be explicitly stated. When discussing AF models, it is preferable to use the term "model" as opposed to the experimentally determined "structures" to avoid any confusion. For AF2 models, it is important to provide the reliability metrics associated with these models (pLDDT and PAE plots) in SI.

vi) R118A DdrO mutation also reduces DdrO-PprI interactions (leading to low cleavage in the absence of ssDNA) as well as interfering with PprI-ssDNA interactions – this should be discussed when analyzing the effects of this mutation.

vii) The proposed model for ssDNA-dependent activation of PprI described in lines 257-258 would be strongly reinforced by performing all-atom molecular dynamics simulations on the ternary complex, which would both validate the ternary complex model and provide important insight into the stabilization of the CSR loop in the cleavage site of PprI.

viii) The biochemical evidence for the existence of a monomer-dimer equilibrium of PprI in solution is very weak, with a very small peak detected by SEC and a very weak band visible after formaldehyde crosslinking. Stronger evidence is needed to claim that PprI exists in an equilibrium between these two states. Alternative approaches may be used to probe this likely low affinity dimer.

ix) Authors suggest that PprI needs to be in a dimer configuration to be stimulated by ssDNA. How would dimerization (either face-to-face or side-by-side) affect DdrO binding? Can the authors build a model of such a complex?

x) In the discussion, the authors suggest that ssDNA resulting from DNA end resection might be responsible for PprI activation (lines 385-387), yet the DNA damage response and in particular the DNA repair enzymes need to be activated first to be able to release ssDNA as a byproduct. Could high doses of irradiation not lead directly to the accumulation of ssDNA fragments as a result of strand breaks before the intervention of DNA repair systems?

Minor points:

a) The resolution and R_{fact}/R_{free} values for the two crystal structures should be reported in the main text (in the results or the Methods section) since the crystallographic table is in SI.

b) Authors state that a catalytic Mn was bound "weakly" in the active site (line 106) – on what basis do the authors claim that the binding is weak?

c) Line 118, authors suggest that the HTH motif can only bind ssDNA since the cavity is only large enough to accommodate ssDNA, but this is assuming that there is no conformational change upon DNA binding.

d) Line 175, authors discuss the C-terminal portion of ssDNA – this should be the 3' end I suppose.

e) I would recommend labelling the ssDNA nucleotides in all structure panels of Fig.3 as in Fig.3a to facilitate the reading of the figures that are shown in different orientation and indicating the 5' and 3' ends would certainly help also.

f) The first two paragraphs of the discussion can be removed or in part transferred to the introduction.

g) Typo line 348: "bound" should be "bind".

h) In the methods section, supplementary Table 3 should be cited when referring to the different DNA and RNA sequences used.

Reviewer #3 (Remarks to the Author):

This is a rather tight-knit story that documents an activation mechanism for the DNA damage response system of *Deinococcus*. The mechanism itself, using ssDNA as the signal, probably is not surprising and echoes the mechanisms used by bacterial SOS systems. However, the mechanism is well documented in this report and represents a significant advance in understanding. The authors establish that PprI binds to ssDNA and that the binding directly activates its proteolytic cleavage of the DdrO transcriptional regulator. Given the unusual capacity of *Deinococcus* species to survive extreme damage, the work should be interesting to many readers. I had only a few and relatively

minor comments.

Please include the size of PprI in the introduction

Figure 1, the apo structure of this PprI, does not add much to what is known and it need not be presented as a main figure. A very similar structure is available.

Lines 123-127. Do the Arg mutants fold properly? CD is used to test this in other mutants.

Fig 2. Some kinetics would be useful. There is no indication of how long the incubation proceeded in the text. One should not need to search through Methods to get such information.

Are DNA concentrations given in terms of molecules or total nucleotides?

Response to reviewers:

We would like to thank you for your careful reading, helpful and positive comments, and constructive suggestions, which have significantly improved the presentation of our manuscript. We have carefully considered all comments and suggestions from the reviewers and revised our manuscript accordingly. In the following section, we summarize our responses to each comment from the reviewers. We believe that our responses have well addressed the concerns of the reviewers.

REVIEWER COMMENTS

Reviewer #1 (Remarks to the Author):

The authors of this manuscript aim to elucidate the mechanism of activation of the protease PprI cleaving DdrO, the transcriptional repressor targeted by this protein in the radioresistant bacterium *Deinococcus geothermalis*. These two proteins play a major role in the expression of DNA repair genes in several *Deinococcaceae* after exposure to irradiation. This manuscript contains numerous experiments and several interesting results, in particular the co-crystallization of PprI with single-stranded DNA, but much information is missing, making it impossible to clearly explain how certain experiments were carried out, and therefore the relevance of several results. Several previous works published on this subject, sometimes performed by other research teams, are not cited and/or discussed according to the results of this manuscript.

Author's response: We are grateful to the reviewer for suggestions and comments for improving the work quality. In the revised version, more experiments including metal ion titration assays, intracellular Mn and Zn concentrations measurements, and RDRM-containing dsDNA binding were performed to enhance our conclusion. Additional experimental information was provided, and previous works were appreciated, discussed, and cited in the revised version.

The authors justify adding MnCl_2 for protein crystallization, as this ion would be preferentially used for DdrO cleavage. However, the manuscript cited as a reference

did not aim to compare different ions for the cleavage efficiency of PprI. In contrast, another manuscript, that is not cited here (Blanchard et al, 2017) highlighted that Zn ions were preferred for the protease activity of PprI from *Deinococcus deserti*. This result is in agreement with the crystallographic data of the *D. deserti* apo protein PprI published in 2009 (Vujicic-Zaggar et al. 2009): PDB 3TDE (inactive apo-enzyme), 3DTI (1 Zn, active enzyme) and 3DTK (2 Zn at the active site, inactive enzyme because too many divalent cations inactivate the enzyme (problem of accessibility of the water molecule as for the thermolysin). The catalytic metal is coordinated by the same HEISH motifs in both PprI proteins (Figure S1) and the two structures are superimposable. The authors do not discuss this important difference even though these two proteins are very similar in sequence and structure. Furthermore, in Blanchard et al, 2017, the authors also showed that metal concentration plays an important role for cleavage efficiency of IrrE and the activation occurred *in vitro* with Zn^{2+} , Mn^{2+} , Fe^{2+} , Co^{2+} but not by Ni^{2+} , Ca^{2+} , Mg^{2+} . They also show that *in vivo* only Zn is capable of activating IrrE. The high concentration used here (2 mM) could have an inhibitory role and explain why no cleavage is observed with Zn (Figure 2C). This is an important point since of all the experiments carried out with the wild-type protein and the numerous mutants were tested with this high concentration of Manganese and not the Zn ion.

Author's response: Thank you for the comments. As you mentioned previous work by Blanchard et al showed that the Zn ion was important for the activation of *D. deserti* PprI and excess zinc inhibited its protease activity. However, despite the Zn observed in the crystal structures of *D. deserti* PprI (PDB 3TDE, 3DTI, and 3DTK), their work also confirmed that Mn^{2+} could stimulate the cleavage of *D. deserti* PprI *in vitro* (Blanchard et al., 2017, Figure 1b), which was consistent with our results. Actually, we have performed cleavage assays in the presence of different concentrations of these two metal ions (1, 5, 20, 100, 500, 1000, 2000, and 4000 μ M, respectively) to investigate the possible inhibitory role of high concentrations of Zn or Mn on DG-PprI cleavage (please see gels below, a and b). While 20 μ M Zn effectively activated DG-PprI cleavage, high concentrations of Zn (more than 100 μ M) strongly inhibited the cleavage

reactions (panel a), which was consistent with previous results shown by Blanchard et al., 2017 and the non-cleavage of DG-PprI in Figure 2c (2 mM). However, 5 μ M Mn was sufficient to stimulate the DG-PprI cleavage, and a minor inhibitory effect was observed only with extremely high Mn concentration (4 mM, panel b). Moreover, we have measured intracellular concentrations of Mn and Zn of *D. geothermalis* and *D. radiodurans* using ICP-MS (data shown below, c), indicating that the intracellular Mn was higher than Zn in both bacteria. It has been well accepted that such high intracellular concentrations of Mn (0.2 to 4 mM) of *D. geothermalis* and *D. radiodurans* played a critical role in their robustness (Daly MJ et al. 2004, Borsetti et al. 2018). Indeed, the ssDNA activation assay titrated with high concentrations of Mn exhibited little inhibitory effect (shown below, d). Thus, the metal ion preference of PprI between *Deinococcus* strains *in vitro* may differ due to varied living environments and intracellular metal concentrations. Given the urgent need for elevated expression of DDR genes by activated PprI and combined with Mn ion observed in PprI-DNA structure, we conclude Mn is preferred for DG-PprI cleavage and 2 mM Mn used in the cleavage assays was reasonable. Nevertheless, zinc ions (Zinc shock) may also play an important role in PprI regulation as reported by Blanchard et al., 2017, which has been appreciated and cited (Page 4, line 81, reference 27) in the revised text.

Reference:

Blanchard L, et al. Conservation and diversity of the IrrE/DdrO-controlled radiation response in

radiation-resistant *Deinococcus* bacteria. *MicrobiologyOpen* **6**, (2017).

Daly MJ, *et al.* Accumulation of Mn(II) in *Deinococcus radiodurans* facilitates gamma-radiation resistance. *Science* **306**, 1025-1028 (2004).

Borsetti F, *et al.* Manganese is a *Deinococcus radiodurans* growth limiting factor in rich culture medium. *Microbiol-Sgm* **164**, 1266-1275 (2018).

- Figure 1C: Several authors of the manuscript already published EMSA assays showing that PprI is able to bind to double-stranded DNA containing an RDR motif (Lu *et al.*, DNA repair 2011). How can they explain the differences between IrrE from *D. deserti*, *D. radiodurans* and *D. geothermalis* with these apo-protein structures data and also thanks to the structure obtained with single-stranded DNA? What are the structural bases for these differences? How do the authors settle these discrepancies?

Author's response: Thank you for the comments. Yes, our previous work (Lu *et al.*, DNA repair 2012) showed that PprI from *D. radiodurans* was able to bind RDRM-containing dsDNA with thousands-fold protein-to-DNA ratios in the reaction (2.5 μ M DR-PprI vs 2 nM DNA). However, the ssDNA binding of DG-PprI was much stronger, which only required a 10-fold protein-to-DNA ratio (1 μ M DG-PprI vs 0.1 μ M ssDNA, Fig. 1b). Indeed, high protein concentrations of DG-PprI (increased from 1 μ M to 200 μ M, data shown below) also exhibited weak RDRM-containing dsDNA (0.1 μ M ssDNA) binding *in vitro*, which is consistent with our previous report. Moreover, despite the similar overall apo-protein structures between PprI from *D. geothermalis* and *D. deserti*, the residues involved in ssDNA binding patches, e.g., patch1 (Arg267), and patch3 (Arg220) are not strictly conserved (supplementary Fig. 1), which may also cause the discrepancies of ssDNA or dsDNA binding affinity of PprI from different species.

Reference:

Lu H, Chen H, Xu G, Shah AM, Hua Y. DNA binding is essential for PprI function in response to radiation damage in *Deinococcus radiodurans*. *DNA repair* **11**, 139-145 (2012).

- Figure 1E No shift is observed in the presence of a short oligonucleotide (10 nt), whereas a stimulation of cleavage is observed in the presence of an oligonucleotide of the same size (Figure 2D): How do the authors settle these two experiments?

Author's response: Yes, our data showed that ssDNA activated DG-PprI cleavage in a length-dependent manner. Thus, despite that short oligonucleotide (10 nt) was unable to form a stable complex with DG-PprI in the EMSA, weak or transient short ssDNA binding of DG-PprI could still activate its cleavage.

- No information is given about the conditions of expression of the wild-type PprI protein, and the different mutated forms of this protein in *D. geothermalis* from the shuttle vector. Did they cloned the genes under the control of the PprI promoter or of another promoter? In particular, the authors indicate an overexpression of these proteins in vivo (Lines 156, 193, 307). Is protein expression from the vector continuous or induced and in the latter case how? Furthermore, the expression kinetics of *ddrO* regulon genes (*recA*, *uvrD*, *ddrO*) under PprI control in the YR1-dg_pprI strain are not comparable with those in wt cells (Figures 3F and 5F). They also differ in vivo (Figure S4C) and between experiments for the same strain (Figure 3E and Figure S4C). The wild-type protein is less effective than the mutants tested (Figure 5E). These differences are not explained by the authors.

Author's response: Thank you for the comments. The shuttle vector pRADK containing *groEL* promoter (strong promoter) was frequently used for continuous protein expression in *D. radiodurans* by people working on this bacterium (Gao G et al. 2005, Helalat SH et al. 2021, Ujaoney AK et al. 2023). In our experiments, the PprI of *D. geothermalis* was heterologously expressed in *D. radiodurans* to assess its functions. Thus, the slight discrepancies of the expression kinetics of *ddrO* regulon genes (Fig. 3f

and 5f) and the phenotypic assays (Fig. 3e and supplementary Fig. 4c) could be explained by the faster cleavage of DdrO *in vivo* (strong induction of PprI protein compared with WT strain). This information was provided in the revised Methods section as following: “To obtain complementary strains for phenotypic assays, PCR amplified wild-type, truncated, and site-mutant *dg_pprI* were cloned into the continuous protein expression shuttle plasmid pRADK containing strong promoter *groEL*^{67,68} and transformed into the *dr_pprI* knock-out strain (YR1).” (revised Page 21, Line 431-435) Moreover, the phenotype assays were repeated (as revised Fig. 3e and 5e, please see below) by optimizing the incubation time after treatments, which exhibited comparable results for the same strain.

Reference:

Gao GJ, Lu HM, Huang LF, Hua YJ. Construction of DNA damage response gene *pprI* function-deficient and function-complementary mutants in *Deinococcus radiodurans*. *Chinese Sci Bull* **50**, 311-316 (2005).
 Helalat SH, Jers C, Behahani M, Mohabatkar H, Mijakovic I. Metabolic engineering of *Deinococcus radiodurans* for pinene production from glycerol. *Microb Cell Fact* **20**, 187 (2021).
 Ujaoney AK, Anaganti N, Padwal MK, Basu B. *Deinococcus* lineage and Rad52 family-related protein DR0041 is involved in DNA protection and compaction. *Int J Biol Macromol* **248**, 125885 (2023).

- The authors did not explain the choice of the housekeeping gene *dr_1343* for qPCR normalization

Author’s response: Thank you for the comment. *dr_1343* is a housekeeping gene showing stable transcriptional levels and suitable for the qPCR normalization, which was frequently used by both our and other groups (eg., Cusick KD et al. 2015, Dai JL et al. 2020). This was mentioned in the revised text “SYBR Premix Ex Taq (Takara, Japan) was used to perform quantitative real-time PCR, and the housekeeping gene *dr_1343* was used for normalization as described previously⁷¹.” (Page 25, Line 512-514)

Reference:

Cusick KD, Fitzgerald LA, Cockrell AL, Biffinger JC. Selection and Evaluation of Reference Genes for Reverse Transcription-Quantitative PCR Expression Studies in a Thermophilic Bacterium Grown under Different Culture Conditions. *PLoS one* **10**, e0131015 (2015).

Dai JL, *et al.* Late embryogenesis abundant group3 protein (DrLEA3) is involved in antioxidation in the extremophilic bacterium *Deinococcus radiodurans*. *Microbiol Res* **240**, (2020).

- Figure 4b: little information is given on the parameters used to build the model: The manuscript presents predicted 3D structures of proteins/complexes that were obtained with AlphaFold2, more details regarding the parameters used for these predictions and an assessment of the quality/confidence levels of these predictions should be given and discussed. This is critical for the comparisons for instance where side chains are compared. What is the confidence in the predicted structures for these regions, and consequently would they impact the interpretation? Could they provide a model with the full DdrO protein and not just with CTD?

Author's response: Thank you for the comments and suggestions. DdrO homodimer was validated in our previous study (Lu *et al.* 2019). For structural prediction, PprI-DdrO (full-length monomeric DdrO) complex was first obtained by AlphaFold2. Five AlphaFold-Multimer models of PprI-DdrO were obtained and ranked according to model confidence (ipTM+pTM) (revised supplementary Table 4, shown below). Then, the top-ranked model was selected for further investigation (revised supplementary Fig. 9, shown below). The pLDDT (per-residue local distance difference test) and PAE (predicted aligned error) plots have also been provided in the supplementary information (revised supplementary Fig. 9, shown below) to cross-verify the convincing prediction of the top-ranked model. Subsequently, PprI-DdrO (full-length DdrO dimer) was obtained by docking the DdrO dimer onto this top-ranked model and the binary model was overlaid onto the PprI-ssDNA structure to generate the ternary model. Moreover, this model was also validated by all-atom molecular dynamics simulations in the revised text (revised Fig. 4b, shown below). The 500 ns simulations showed that the PprI-DdrO-ssDNA ternary complex is convincing and the RMSD

values (less than 2 Å) during MD simulations demonstrating the stability of the CSR loop in the cleavage site of PprI (revised shown in Fig. 4d, shown below). It has been revised as following: “Thus, to obtain possible structural information of ssDNA-mediated activation of DdrO cleavage by PprI, AlphaFold2 complex modeling tool⁴¹,⁴² was used to predict the model of PprI binding to the C-terminal region of DdrO (full-length monomeric DdrO, Supplementary Table 4, Supplementary Fig. 9) and then DdrO dimer (Protein Data Bank [PDB] ID: 6JQ1)²⁵ was docked onto the predicted complex (binary complex), followed by overlaying this model on the PprI-ssDNA structure to obtain a ternary model (Fig. 4b). Furthermore, all-atom molecular dynamics (MD) simulations were performed to validate the ternary complex models, following similar protocols in our previous studies on the protein dynamics^{43, 44, 45, 46, 47}.” (revised Page 12, line 243-251)

ranked	ipTM+pTM
1	0.799
2	0.557
3	0.496
4	0.478
5	0.223

Fig. s9

Fig. 4b

Fig. 4d

Reference:

Lu H, *et al.* Structure and DNA damage-dependent derepression mechanism for the XRE family member

- Two different organizations of the PprI protein were observed in the crystals obtained with the apo-protein and the protein bound to the single-stranded oligonucleotide (side by side and face to face respectively), but only a very small number of protein dimers were observed in solution even after attempting to crosslink the monomers. No change was observed when ssDNA was added (Figure 5 and Figure S7A and b). The mutations introduced at specific locations (Patch 1 and 2) did not decrease the cleavage of ddrO, excepting if 6 different mutations were introduced at specific positions which would impair the equilibrium monomer/dimer of PprI. What conclusion can be drawn from these results? What are the consequence of a PprI dimerization on the model of the interaction between DdrO and PprI (Figure 4b)?

Author's response: Yes, the 6mut destroying both NTD interfaces impaired the monomer/dimer equilibrium of PprI in solution (Fig. 5c) as well as the radioresistance of the hexa-mutant complementary strain (YR1-6mut, Fig. 5e), which indicated that dimer formation is important for the inherent protease activity of PprI (as mentioned in Page 15, line 321-334). Given that the NTD of PprI containing the catalytic center also recognizes the CSR loop of DdrO, we suspect that PprI dimerization may facilitate DdrO interactions compared with monomeric PprI. Unfortunately, both AF2 and manual building failed to further obtain the PprI-DdrO (dimer-dimer) model because of the complex interactions between these two dimers (steric clashes) that large conformational changes may be required. More evidence is needed (e.g., the structure of dimeric PprI in complexed with DdrO dimer) to draw the detailed mechanism.

- The model proposed by the authors about the role of single strand DNA in vivo may be questionable. The authors repeatedly showed that single-stranded DNA is not required for complete cleavage of DdrO by PprI when the cleavage assay was performed at a respective concentration of the two partners of 8/1. Stimulation is observed when the ratio is 8/0.1. In vivo, the PprI protein is reputed to be constitutively expressed, whereas the DdrO expression is repressed by the protein itself. After

irradiation, DdrO is cleaved by PprI, but continued expression of the DdrO protein leads cells to an apoptotic pathway (Devigne et al, 2015). Consequently, why ssDNA would be used to activate PprI if the concentration of PprI into the cells remains higher than that of DdrO *in vivo*?

Author's response: Thank you for the comments. The ratio 8/1 of the two partners was used for the cleavage assay because of the weak protease activity of PprI in the absence of ssDNA *in vitro* (Fig. 2a), even though the abundance of PprI and DdrO *in vivo* remained unclear. Despite the constitutive protein expression in cells as you mentioned, DdrO cleavage by PprI *per se* is supposed to be at a very low level to control the balance of intracellular amount of DdrO as well as the expression of DDR genes under normal conditions. Thus, the activation mechanism is the key to the proper function of the PprI-DdrO system. Interestingly, it has been reported that PprI mainly degrades free DdrO (DNA-unbound DdrO) to regulate the proportion of DNA-bound DdrO *in vivo* (equilibrium between free and DNA-bound DdrO *in vivo*). In our model, the pool of free DdrO dramatically decreases upon ssDNA accumulation (ssDNA is the by-product of large-scale DNA damage), causing the dissociation of DdrO from the promoter and further leading to the upregulation of DDR genes. Thus, ssDNA-activated PprI guarantees the prompt DNA damage response in *D. geothermalis*, which is an alternative pathway for ssDNA-activated DNA damage response in bacteria (e.g., SOS response).

Thank you again for the above suggestions and comments.

Reviewer #2 (Remarks to the Author):

In this study, Lu et al. shed light on the mechanisms underlying the response of the radiation resistant *Deinococcus* bacteria to DNA damage. In these bacteria, the classical SOS response system involving notably LexA and RecA is missing and is instead replaced by a metalloprotease, PprI/IrrE, that cleaves a transcriptional repressor, DdrO, in response to DNA damage. Several articles in the recent years have unveiled the molecular details underlying this process, but so far little is known regarding the mechanisms responsible for activation of PprI/IrrE. Indeed, unlike LexA, PprI/IrrE is constitutively expressed and thus needs to be activated to perform its cleavage of DdrO. Here, the authors report the crystal structure of apo-PprI from *D. geothermalis*, which is very similar to the previously reported structure of *D. deserti* PprI, but more importantly, they reveal that ssDNA can strongly activate the protease activity of PprI and report the crystal structure of ssDNA-bound PprI. This structural and biochemical study, supported by in vivo mutagenesis analysis, represents an important advance in our understanding of the DNA damage response in *Deinococcus* bacteria, identifying ssDNA, which is a by-product of large-scale DNA damage, as a key activator of the response pathway. Overall, the manuscript is very well written and clear, and the presented experiments have been well performed and support the conclusions.

Author's response: We are grateful to the reviewer for appreciating the work and suggestions for improving the work quality.

A few points listed below should nonetheless be addressed before this manuscript can be published.

Major points:

i) The quality of the structures, and in particular the geometry, could certainly be improved to reduce the clashscore and sidechain outliers (and lower the Rfree). For the PprI-ssDNA structure, a more detailed description of the visible DNA bases and their fit to the density and contacts with the protein should be provided – could the authors clearly define the polarity of the two fragments?

Author's response: Thank you for the comments. We have further refined the structures to improve the geometry (clashscore), RSRZ outlier, R.m.s deviations for bond angles, and Rfree values (e.g., for PprI-DNA complex the clashscore decreased from 8 to 4; the RSRZ outliers decreased from 15.1% to 10.4%). The statistics have been updated in the revised Supplementary Table 1 (revised version of validation reports). Yes, bases of dG(7) and dT(11) have very weak electron density in our PprI-ssDNA structure, which has been mentioned in the revised text. “Notably, despite the weak electron density of bases of dG(7) and dT(11), dG(10) exhibited well-defined electron density with its guanine base forming π -stacking interactions with the phenylalanine residue (Phe88, Fig. 3c).” (Page 9, line 185). However, we could clearly see the continuous electron density of the phosphate backbone of ssDNA (Fig. 3c). Moreover, two neighboring purines at 9 and 10 positions (AG) of patch 1 have well-defined electron densities (“patch 1” of Fig. 3c), which indicates the polarity of the two fragments. An additional schematic showing the electron density of all bases and ssDNA-PprI interactions was provided as the revised supplementary Fig. 6 (please see below).

ii) ssDNA used for EMSAs and protease assays should ideally be polydT to avoid the formation of secondary structure, hairpins or partial duplexes. Authors should comment on this and provide evidence that their ssDNA fragments were indeed fully single-

stranded.

Author's response: Thank you for the comments and suggestions. For the ssDNA (35 nt) used for EMSAs and protease assays, the free Gibbs energy (ΔG) values (calculated by OligoAnalyzer Tool from IDT: <https://sg.idtdna.com/calc/analyzer>) of any self-dimers and hairpins are weaker (more positive) than -9 kcal/mol (shown below). These values are comparable to oligos used for co-crystallization (29 nt, -8.5 kcal/mol for self-dimer), which are supposed not to form secondary structures. Moreover, ssDNA binding and activation assays using 35 nt polydT (please see below, a and b) showed the same results as those in Fig. 1b and 2a, which also confirmed the ssDNA property of oligos used in our assays.

	Sequence (5'→3')	ΔG (kcal/mol)	hairpin	self-dimer
35nt	CGCTCTTCGCCATTCTCTTGAAGTTTCAAACCTGG		-2.3	-8.8
29nt	TCATGAGCAGTTTTTTTTTTTTTTTTTTTT		2.2	-8.5

iii) Timecourse experiments to follow the kinetics of the protease activity of PprI would be more insightful than single time point experiments, especially for comparing mutants.

Author's response: Thank you for the good suggestions. Additional timecourse experiments (2, 5, 10, 15, 20, 30, 45 min) were performed and provided in the revised version (revised Supplementary Fig. 8, please see below). Compared with the WT protein, the cleavage of patch2 and 3 mutants (panel c and d) were slightly diminished, while the patch1 and 6mut mutants (panel b and e) exhibited severely impaired cleavage efficiency, which is consistent with the single time point experiments of these mutants

(Fig. 3d, lanes 3, 4, and 5, Fig. 5d, lane 5).

iv) The second ssDNA fragment bound to patch 3 of PprI is actually located at the interface with a neighboring symmetry-related molecule in the crystal – could this be a crystal packing artefact? This should perhaps be discussed in view that mutations in this patch had little effects on the protease activation.

Author's response: Thank you for the comments. The patch3 residues interact with not only the phosphate backbone (e.g., Arg220, Arg250) but also the sugar and base (e.g., Arg253, Met255) of ssDNA, which undergo rotamer changes compared with the apo-protein structure (please see figure below, white: apo-protein, purple: PprI-ssDNA complex, black arrows indicate the rotamer changes). Moreover, given the abolished ssDNA binding (Fig. 3c) and partially impaired activation (Fig. 3d, lane 13 and revised Supplementary Fig. 8d) of patch3 mutant, we concluded that it does bind ssDNA and exclude the possible artifact of the crystal packing forces.

v) Authors state that they used AlphaFold2 to model the ternary structure of PprI-DdrO-ssDNA, but this is incorrect. They used AF2 to model PprI binding to the C-terminal region of DdrO and then overlaid this model on the PprI-ssDNA structure to build a ternary model. This should be explicitly stated. When discussing AF models, it is preferable to use the term “model” as opposed to the experimentally determined “structures” to avoid any confusion. For AF2 models, it is important to provide the reliability metrics associated with these models (pLDDT and PAE plots) in SI.

Author’s response: Thank you very much for the comments and suggestions. The “structures” were rephrased as “model” in the revised text and pLDDT and PAE plots were provided in the revised supplementary Fig. 9 to verify the quality/confidence levels of this model (please see below), suggesting the convincing prediction. Moreover, the method for obtaining the AF2 model was updated in the revised text as following: “Thus, to obtain possible structural information of ssDNA-mediated activation of DdrO cleavage by PprI, AlphaFold2 complex modeling tool^{41, 42} was used to predict the model of PprI binding to the C-terminal region of DdrO (full-length monomeric DdrO, Supplementary Table 4, Supplementary Fig. 9) and then DdrO dimer (Protein Data Bank [PDB] ID: 6JQ1)²⁵ was docked onto the predicted complex (binary complex), followed by overlaying this model on the PprI-ssDNA structure to obtain a ternary model (Fig. 4b).” (Page 12, line 243-251)

Fig. s9

vi) R118A DdrO mutation also reduces DdrO-PprI interactions (leading to low cleavage in the absence of ssDNA) as well as interfering with PprI-ssDNA interactions – this should be discussed when analyzing the effects of this mutation.

vii) The proposed model for ssDNA-dependent activation of PprI described in lines 257-258 would be strongly reinforced by performing all-atom molecular dynamics simulations on the ternary complex, which would both validate the ternary complex model and provide important insight into the stabilization of the CSR loop in the cleavage site of PprI.

Author's response: Thank you for these two comments and very insightful suggestions. All-atom molecular dynamics simulations have been conducted to validate the model of the predicted ternary complex. The RMSD values (less than 2 Å) during the 500 ns MD simulations demonstrated the stability of the CSR loop in the cleavage site of PprI (as revised Fig. 4d, please see below). Moreover, the simulation results also indicated that Arg118 directly interacts with the phosphate group of ssDNA in the presence of ssDNA, which was consistent with FRET assays (Fig. 4a) and the relative binding free energy changes ($\Delta\Delta G$) of R118A mutation in the presence of ssDNA (revised Fig. 4e, please see below).

It has been discussed and revised as following. “Furthermore, all-atom molecular dynamics (MD) simulations were performed to validate the ternary complex models, following similar protocols in our previous studies on the protein dynamics^{43, 44, 45, 46, 47.}” (Page 12, Line 250-252)

“It has been previously reported that the cleavage site region (CSR) loop (residues 116-

121 in DG-DdrO) on the C-terminal domain of DdrO is critical for the specific cleavage by PprI²⁴. In the complex model, the CSR loop interacts with the N-terminal zinc peptidase-like domain of PprI by forming anti-parallel β -strands (Fig. 4b, c), and its binding stability is verified by the RMSD values (less than 2 Å) during MD simulations (Fig. 4d). Notably, the CSR loop protrudes into the cleavage site of PprI and Arg118 of the CSR loop directly interacts with the phosphate group of the ssDNA in the predicted PprI-DdrO-ssDNA model (Fig. 4b, c). Such interactions were consistent with subsequent FRET assay results using R118A mutant DdrO, which exhibited lower interaction between PprI and DdrO compared to wildtype PprI and non-enhanced PprI-DdrO interactions in the presence of ssDNA (Fig. 4a). In addition, we performed *in silico* mutagenesis (free energy perturbation calculations (FEP)^{48, 49, 50} for R118A) to estimate the energy contribution of R118 to the binding of DdrO to PprI in the ternary model. The FEP results of R118A mutation ($\Delta\Delta G = 5.22 \pm 0.33$ kcal/mol in the presence of ssDNA) showed that the mutation would decrease the interaction between PprI and DdrO, providing explanations for the FRET assay results (Fig. 4a, e). Moreover, DdrO mutant proteins bearing alanine substitutions of key CSR residues (E116A, L117A, R118A, and G119A) were purified and subjected to PprI cleavage and ssDNA activation assays to further confirm possible interactions between the CSR loop and ssDNA (Fig. 4f). In the absence of ssDNA, all the DdrO mutants except L117A exhibited considerably decreased cleavage by PprI (1 μ M, lanes 1-10), suggesting that these residues are involved in the interaction.” (Page 12, Line 253-274)

viii) The biochemical evidence for the existence of a monomer-dimer equilibrium of

PprI in solution is very weak, with a very small peak detected by SEC and a very weak band visible after formaldehyde crosslinking. Stronger evidence is needed to claim that PprI exists in an equilibrium between these two states. Alternative approaches may be used to probe this likely low affinity dimer.

Author's response: Thank you for the comments and suggestions. It is very difficult to probe this low-affinity dimer because of the limitation of technical conditions in our lab. However, we conducted additional FRET assays in the revised version to detect the weak and dynamic dimer formation of PprI (revised supplementary Fig. 10e, please see below). The results showed that mutations of the two interfaces (2mut or 4mut), especially the double interface mutation (6mut) led to a significant decrease in the ratio RFU 530/480 compared with the WT protein, which indicated the very weak dimerization of PprI in solution. It has been revised as following: "In contrast, the hexa-mutant (6mut) destroying both interfaces showed an imperceptible peak corresponding to the DG-PprI dimer (Fig. 5c) as well as significant decreased FRET signals (Supplementary Fig. 10e), indicating the coexistence of both interfaces and dynamic behavior of the monomer-dimer equilibrium of DG-PprI in solution." (Page 15, Line 315-319)

ix) Authors suggest that PprI needs to be in a dimer configuration to be stimulated by ssDNA. How would dimerization (either face-to-face or side-by-side) affect DdrO binding? Can the authors build a model of such a complex?

Author's response: Thank you for the comments and suggestions. Given that the NTD

of PprI recognizes the CSR loop of DdrO and our current experimental data, we suspect that PprI dimerization may facilitate DdrO interactions compared to the monomeric PprI. However, more evidence is required to illustrate the detailed mechanism. We have also tried to build the model of dimeric PprI in complex with DdrO dimer. Unfortunately, both AF2 and manual building failed to obtain the PprI-DdrO (dimer-dimer) model because of the complex interactions between these two dimers (steric clashes) that large conformational changes may be required.

x) In the discussion, the authors suggest that ssDNA resulting from DNA end resection might be responsible for PprI activation (lines 385-387), yet the DNA damage response and in particular the DNA repair enzymes need to be activated first to be able to release ssDNA as a byproduct. Could high doses of irradiation not lead directly to the accumulation of ssDNA fragments as a result of strand breaks before the intervention of DNA repair systems?

Author's response: Thank you for the comments. To date, in the bacterial DNA repair field, it is generally accepted that ssDNA is one of the important byproducts of large-scale DNA damage, which triggers DNA damage response such as the well-known SOS response in *E. coli*. Indeed, it has also been suggested that the damage to the genome could be indirect effects such as radiation-induced reactive oxygen species (ROS) surrounding the DNA, which may also activate the intracellular DNA repair system to a certain extent. However, given the tolerance and the prompt DNA damage response of *Deinococcus* species under extremely high doses of irradiation, the ssDNA directly generated by irradiation is supposed to be one of the most important signals triggering the cellular DNA repair.

Minor points:

a) The resolution and R_{fact}/R_{free} values for the two crystal structures should be reported in the main text (in the results or the Methods section) since the crystallographic table is in SI.

Author's response: Thank you for the suggestion. The revised Rwork/Rfree values and resolution were provided in the revised manuscript as following: "We first determined the crystal structure of full-length DG-PprI at the resolution of 2.8 Å (Rwork/Rfree=0.247/0.284) in the presence of Mn²⁺" (Page 6, Line 102-103)

"crystals of the PprI-ssDNA complex containing 29nt ssDNA were grown in the presence of Mn²⁺ ions and diffracted X-rays to ~4Å and subsequently improved to 2.2Å (Rwork/Rfree=0.233/0.265)" (Page 9, Line 171-174)

b) Authors state that a catalytic Mn was bound "weakly" in the active site (line 106) – on what basis do the authors claim that the binding is weak?

Author's response: Yes, the Mn ions are present in both apo and ssDNA-bound PprI structures. However, the electron density of Mn in the apo structure is weak compared with this of PprI-ssDNA structure due to the lack of interactions with residues from neighboring PprI molecule. It has been revised as following: "Despite weak electron density in apo structure, a catalytic manganese ion (Mn²⁺) lies in the active site of DG-PprI, coordinated by the conserved HEXXH motif (HEISH in DG-PprI) of the N-terminal zinc peptidase-like domain (Fig. 1a)." (Page 6, Line 113-115)

c) Line 118, authors suggest that the HTH motif can only bind ssDNA since the cavity is only large enough to accommodate ssDNA, but this is assuming that there is no conformational change upon DNA binding.

Author's response: Thank you for the comments. Based on our PprI-ssDNA structure, the partially buried HTH motif appears to be suitable for ssDNA binding (Supplementary Fig. 2), which was further confirmed by EMSA assays (Fig. 1b). In the revised text, we rephrased this sentence by deleting the word "only": "In contrast to the solvent-exposed HTH motifs of well-documented transcription factors (e.g., Lex A and DdrO protein^{33, 34}), the HTH motif of DG-PprI is partially buried and capped by its N-terminal domain, resulting in a space suitable for ssDNA access (Fig. 1a and Supplementary Fig. 2)." (Page 7, Line 122-125)

d) Line 175, authors discuss the C-terminal portion of ssDNA – this should be the 3' end I suppose.

Author's response: It has been corrected as "3'-TTTTT", thank you. (Page 9, Line 184)

e) I would recommend labelling the ssDNA nucleotides in all structure panels of Fig.3 as in Fig.3a to facilitate the reading of the figures that are shown in different orientation and indicating the 5' and 3' ends would certainly help also.

Author's response: Thank you for the suggestions. The ssDNA nucleotides and the orientation of the 5' and 3' ends were labeled in the revised Fig.3 (please see below).

f) The first two paragraphs of the discussion can be removed or in part transferred to the introduction.

Author's response: We have revised and transferred the partial discussion of the first two paragraphs to the introduction to make it more compact. Thank you.

g) Typo line 348: “bound” should be “bind”.

Author’s response: Thank you, it has been corrected. (Page 17, Line 346)

h) In the methods section, supplementary Table 3 should be cited when referring to the different DNA and RNA sequences used.

Author’s response: Thank you. It has been cited in the methods section when referring to the different DNA and RNA sequences used. (Page 23, Line 457-458; Page 24, Line 483-484, Line 490-491; Page 25, Line 504-505, Line 514-515)

Thank you again for your careful reading, helpful comments, and constructive suggestions, which have significantly improved the presentation of our manuscript.

Reviewer #3 (Remarks to the Author):

This is a rather tight-knit story that documents an activation mechanism for the DNA damage response system of *Deinococcus*. The mechanism itself, using ssDNA as the signal, probably is not surprising and echoes the mechanisms used by bacterial SOS systems. However, the mechanism is well documented in this report and represents a significant advance in understanding. The authors establish that PprI binds to ssDNA and that the binding directly activates its proteolytic cleavage of the DdrO transcriptional regulator. Given the unusual capacity of *Deinococcus* species to survive extreme damage, the work should be interesting to many readers. I had only a few and relatively minor comments.

Author's response: We are grateful to the reviewer for appreciating the work and suggestions for improving the work quality.

Please include the size of PprI in the introduction

Author's response: Yes, it has been mentioned in the introduction section. "The distinct DNA damage response pathway mediated by the metallopeptidase PprI (31.3 kDa for DG-PprI) and the transcription repressor protein DdrO (15.7 kDa for DG-DdrO) has been characterized and extensively studied in *Deinococcus* in recent years". (Page 4, Line 64-66)

Figure 1, the apo structure of this PprI, does not add much to what is known and it need not be presented as a main figure. A very similar structure is available.

Author's response: Thank you for the suggestion. The apo structure of PprI (Fig. 1a) has been moved to the revised Supplementary Fig. 2a.

Lines 123-127. Do the Arg mutants fold properly? CD is used to test this in other mutants.

Author's response: Thank you for the comment and suggestion. In the revised version, additional CD assays have been conducted to confirm the proper folding of the Arg

mutants, please see the results below (revised Supplementary Fig. 10d).

Fig. s10d

Fig 2. Some kinetics would be useful. There is no indication of how long the incubation proceeded in the text. One should not need to search through Methods to get such information.

Author's response: Thank you. The incubation time has been provided in the revised figure legends as following. "Activation assays showing the ssDNA-enhanced PprI cleavage. DG-DdrO (8 μ M) was incubated with DG-PprI (0.1 μ M) in the presence of 2 mM MnCl₂ in the absence or presence of 35nt ssDNA (0.1 μ M) at 37 °C for 30 minutes." (Page 36, Line 812-814) Moreover, we have performed timecourse cleavage assays to obtain the kinetics of mutant proteins (2, 5, 10, 15, 20, 30, 45 min, as revised Supplementary Fig. 8, please see below). Compared with the WT protein, the cleavage of patch2 and 3 mutants (panel c and d) were slightly diminished, while the patch1 and 6mut mutants (panel b and e) exhibited severely impaired cleavage efficiency, which is consistent with the single time point experiments of these mutants (Fig. 3d, lanes 3, 4, and 5, Fig. 5d, lane 5).

Are DNA concentrations given in terms of molecules or total nucleotides?

Author's response: Yes, they are given in terms of molecules (molar concentration measurement unit).

Thank you again for your helpful comments and constructive suggestions.

Reviewer #1 (Remarks to the Author):

In this revised form of the manuscript and in the response to the reviewers, the authors show the results of various experiments which allow a better understanding of several experiments submitted in the first version and improve the quality of the manuscript. They also answer a number of questions I had addressed. Nevertheless, after reading this revised paper, I still have comments:

1-It remains difficult to directly connect the results obtained in vivo with those performed in vitro. The authors showed here that in vitro DdrO cleavage by PprI at a low molar ratio is stimulated by the presence of ssDNA while at a higher molar ratio a complete cleavage is observed. In vivo the gene encoding PprI is reputed to be continuously expressed. It is unfortunate that the expression of wild-type or mutated proteins was not carried out using the natural promoter of the pprI gene. The promoter used in this study (GroEL) certainly enables continuous expression of the genes under its control, but is pGroEL as effective than the natural pprI promoter? The new figures 3E and 5E seem now to indicate that this is true (not in figure S4, for which survival after irradiation remains 2 log more sensitive than in the wild-type strain when the gene is expressed from the plasmid). Moreover, a delay of 30-45 minutes is observed to reach the maximal expression of recA and UvrD when compared to the wt strain suggesting, as proposed in the responses to the authors, a faster cleavage of DdrO in the wt strain. Is it due to a variation of the amount of pprI/irrE produced when expressed from the vector when compared to the wt strain? a difference of protein stability in vivo of the mutant proteins? It was previously shown that the start codon position of *D. radiodurans* irrE has been wrongly predicted. The purified protein containing 40 extra residues, may have influenced its properties and therefore the results of previous experiments. How the authors cloned here the *D. geothermalis* wt or the different versions of pprI mutant genes?

3- It is also important for future readers of this manuscript that several results presented in the responses to the reviewers will also be published, at least as supplementary data. If some of them have already been published in the past, they should be referenced. For example, the experiments in DR and DG showing the different effect of zinc and manganese on the cleavage efficiency of ddrO by PprI in vitro, as well as the different concentrations of these two metals in the cell in vivo, should be associated with this publication or referenced. Reference 25 in the main text concerning the preferential use of Manganese is not appropriate as in this paper no comparison between the different metals has been performed contrary to the gels shown in the reply to the reviewers.

4-It is also important that several of these results are better discussed in the main text. In the model proposed by Blanchard et al, 2017, they did not observe binding of the *D. deserti* IrrE to RDRM-containing DNA fragments or to DdrO-bound DNA. Concerning metal specificity and DNA binding, they also indicated that their results obtained with *D. deserti* IrrE are different from those published previously for *D. radiodurans* IrrE. In their model only a Zinc shock induces PprI/IrrE-dependent DdrO cleavage in vivo in *D. deserti*. The data presented here showed that the cleavage efficiency of PprI is broadly the same in vitro in the presence of either low or high concentrations of manganese ions, and since the concentration of manganese ions is naturally high in *D. geothermalis*, why is DdrO not always cleaved in vivo? Are the authors suggesting that in DG, contrary to what has been shown in *D. deserti*, the presence of single-stranded DNA is essential for IrrE to cleave DdrO? If so, do the results presented here by the authors are specific of what happen in *D. geothermalis* or can be generalized to other *Deinococcus* species? This is not a detail point, because as the authors point out in the introduction, several hypotheses have been put forward in the past to explain the activation mechanism of the pprI/IrrE metalloprotease, a key point in these species for the response to irradiation.

Reviewer #2 (Remarks to the Author):

The authors have fully addressed my comments and have significantly improved and strengthened their manuscript by performing some important additional experiments. I now support the publication of this work.

Reviewer #3 (Remarks to the Author):

I think the authors have done a good job addressing referee comments and I am satisfied with the result. It is a good revision and an interesting paper.

Response to reviewers:

We would like to thank Reviewer #2 and #3 for their appreciation of the revision. We have carefully considered all the comments and suggestions from Reviewer #1 and revised our manuscript accordingly. We believe that our responses have addressed the comments by Reviewer #1.

Reviewer #1 (Remarks to the Author):

In this revised form of the manuscript and in the response to the reviewers, the authors show the results of various experiments which allow a better understanding of several experiments submitted in the first version and improve the quality of the manuscript. They also answer a number of questions I had addressed. Nevertheless, after reading this revised paper, I still have comments:

Author's response: Thank you for appreciating the first revision and we have further revised the MS according to the comments you mentioned.

1. It remains difficult to directly connect the results obtained in vivo with those performed in vitro. The authors showed here that in vitro DdrO cleavage by PprI at a low molar ratio is stimulated by the presence of ssDNA while at a higher molar ratio a complete cleavage is observed. In vivo the gene encoding PprI is reputed to be continuously expressed. It is unfortunate that the expression of wild-type or mutated proteins was not carried out using the natural promoter of the pprI gene. The promoter used in this study (GroEL) certainly enables continuous expression of the genes under its control, but is pGroEL as effective than the natural pprI promoter? The new figures 3E and 5E seem now to indicate that this is true (not in figure S4, for which survival after irradiation remains 2 log more sensitive than in the wild-type strain when the gene is expressed from the plasmid). Moreover, a delay of 30-45 minutes is observed to reach the maximal expression of recA and UvrD when compared to the wt strain suggesting, as proposed in the responses to the authors, a faster cleavage of DdrO in the wt strain. Is it due to a variation of the amount of pprI/irrE produced when expressed from the vector when compared to the wt strain? a difference of protein stability in vivo of the

mutant proteins?

Author's response: Thank you for the comments. The pRADK vector is a canonical plasmid for endogenous gene and homolog gene complementary assays in the field of *D. radiodurans* study, which has been widely used not only by our group but also by other groups (Hua et al., 2003, Lecointe et al., 2004, Ohba et al., 2005, Gao et al., 2006, Misra et al., 2006, Satoh et al., 2009). Moreover, we have compared the transcriptional levels of *pprI* in both wild-type *D. radiodurans* strain (under the control of natural *pprI* promoter) and *YR1-dg_pprI* complementary strain (under the control of GroEL promoter), which suggested the comparable expression levels (no significance using T-test, consistent with the transcriptional analysis in references: Lecointe et al., 2004, please see figure below). Thus, the discrepancy you mentioned could be due to the slight variations of the intrinsic properties between DR-PprI and DG-PprI. However, it is difficult to test the protein properties (e.g., stability) *in vivo*. Nevertheless, these experiments were done to provide the *in vivo* evidence for the structural observed features (mutants), which support our conclusion.

Reference:

Hua, Y. et al. PprI: a general switch responsible for extreme radioresistance of *Deinococcus radiodurans*. *Biochem Biophys Res Commun* 306, 354-60 (2003).

Lecointe, F., Coste, G., Sommer, S. & Ballone, A. Vectors for regulated gene expression in the radioresistant bacterium *Deinococcus radiodurans*. *Gene* 336, 25-35 (2004).

Ohba, H., Satoh, K., Yanagisawa, T. & Narumi, I. The radiation responsive promoter of the *Deinococcus radiodurans pprA* gene. *Gene* 363, 133-41 (2005).

Gao, G. et al. Internal promoter characterization and expression of the *Deinococcus radiodurans* pprI-folP gene cluster. FEMS Microbiol Lett 257, 195-201 (2006).

Misra, H.S. et al. An exonuclease I-sensitive DNA repair pathway in *Deinococcus radiodurans*: a major determinant of radiation resistance. Mol Microbiol 59, 1308-16 (2006).

Satoh, K., Tu, Z., Ohba, H. & Narumi, I. Development of versatile shuttle vectors for *Deinococcus grandis*. Plasmid 62, 1-9 (2009).

2. It was previously shown that the start codon position of *D. radiodurans* irrE has been wrongly predicted. The purified protein containing 40 extra residues, may have influenced its properties and therefore the results of previous experiments. How the authors cloned here the *D. geothermalis* wt or the different versions of pprI mutant genes?

Author's response: Yes, the start codon position of *D. radiodurans* pprI has been wrongly predicted previously. However, the DG-PprI does not have such extra residues, which could be aligned well with the corrected *D. radiodurans* PprI (without 40 extra residues).

3. It is also important for future readers of this manuscript that several results presented in the responses to the reviewers will also be published, at least as supplementary data. If some of them have already been published in the past, they should be referenced. For example, the experiments in DR and DG showing the different effect of zinc and manganese on the cleavage efficiency of ddrO by PprI *in vitro*, as well as the different concentrations of these two metals in the cell *in vivo*, should be associated with this publication or referenced.

Author's response: Thank you for the comments and suggestions. As per your comments, several results presented in the previous responses have been referenced or added to supplementary data. For example, the experiments showing the different effects of zinc and manganese on the cleavage efficiency of DdrO by PprI *in vitro*, as well as the different concentrations of these two metals in the cell *in vivo* (Supplementary Fig. 11, shown below). It has been revised as following: “Compared to radiosensitive bacteria such as *E. coli*, *Deinococcus* species have much higher

intracellular Mn concentrations (ranging from 0.2 to 4 mM, Supplementary Fig. 11a) as well as the manganese-to-iron ratio (0.24 in *D. radiodurans* and 0.46 in *D. geothermalis*)^{53, 54}, which contribute to the proteome protection critical for not only antioxidation but also DNA repair. Indeed, despite low concentrations of Zn or Mn ions (20 μM) being able to activate the DG-PprI digestion *in vitro*, Mn was effective at high concentrations (Supplementary Fig. 11b,c,d).” (Page 17, line 354-360)

Reference 25 in the main text concerning the preferential use of Manganese is not appropriate as in this paper no comparison between the different metals has been performed contrary to the gels shown in the reply to the reviewers.

Author’s response: It has been removed from the text and the phrase “preferred” has been revised to “effective” for a more accurate description. “We first determined the crystal structure of full-length DG-PprI at the resolution of 2.8 Å (Rwork/Rfree=0.247/0.284) in the presence of Mn²⁺, which is effective for its protease activity.” (Page 4, line 101-103)

4-It is also important that several of these results are better discussed in the main text. In the model proposed by Blanchard et al, 2017, they did not observe binding of the *D. deserti* IrrE to RDRM-containing DNA fragments or to DdrO-bound DNA. Concerning metal specificity and DNA binding, they also indicated that their results obtained with *D. deserti* IrrE are different from those published previously for *D. radiodurans* IrrE. In

their model only a Zinc shock induces PprI/IrrE-dependent DdrO cleavage in vivo in *D. deserti*. The data presented here showed that the cleavage efficiency of PprI is broadly the same in vitro in the presence of either low or high concentrations of manganese ions, and since the concentration of manganese ions is naturally high in *D. geothermalis*, why is DdrO not always cleaved in vivo? Are the authors suggesting that in DG, contrary to what has been shown in *D. deserti*, the presence of single-stranded DNA is essential for IrrE to cleave DdrO? If so, do the results presented here by the authors are specific of what happen in *D. geothermalis* or can be generalized to other *Deinococcus* species? This is not a detail point, because as the authors point out in the introduction, several hypotheses have been put forward in the past to explain the activation mechanism of the pprI/IrrE metalloprotease, a key point in these species for the response to irradiation.

Author's response: Thank you for the comments. Just as you mentioned, there are several hypotheses proposed for the activation mechanism of PprI-DdrO system. Actually, the ssDNA activation mechanism unveiled in this study has no conflict with the former model proposed by Blanchard et al, 2017 that Zinc shock may also induce PprI-dependent DdrO cleavage. Under normal conditions, DdrO exists in both free and promoter-bound forms with the basal activity of PprI in the absence of ssDNA. After DNA damage, the ssDNA activates PprI cleavage (breaking the balance of free DdrO and DNA-bounded DdrO), leading to the upregulation of DDR genes, similar to the SOS response mediated by LexA autocleavage. Given that ssDNA is a direct DNA damage signal and overall conserved protein structures and biochemical properties of PprI-DdrO, we suspect that this mechanism appears to be universal in *Deinococcus* species, or at least in *D. geothermalis* and *D. radiodurans*.

Reviewer #1 (Remarks to the Author):

After reading the responses from the authors of this second review, I agree with the publication of this manuscript.